# Beyond Imitation: A Framework and Benchmark for LLM-Assisted Peer Review

## Abstract

The rapid growth of scientific publishing has strained peer review, particularly in machine learning, raising concerns about declining review quality and increasing reviewer workload. Large language models (LLMs) have been proposed as automated review assistants, yet their evaluation has focused largely on imitating human-written reviews rather than supporting the core functions of peer review. Here, we introduce a verification-centric perspective on LLM-assisted peer review, emphasizing error detection as a critical and resource-intensive task. We present a scalable benchmark that evaluates review systems' ability to identify logical contradictions, constructed through synthetic insertion of errors into conference papers—yielding unambiguous evaluation targets and enabling systematic comparison. We further propose a Multi-Layered Review (MLR) framework that prioritizes detailed manuscript comprehension before review generation, aligning more closely with human reviewing practices while improving token efficiency. Across evaluations, our approach demonstrates strong alignment with human review scores, achieves high error detection performance, and provides complementary perspectives on reviewer focus. These improvements can be attributed to both the choice of the underlying LLM and the design of our system. At the same time, we corroborate persistent vulnerabilities to adversarial manipulation, underscoring the need for robustness in automated review systems. Our findings highlight the importance of rigorous, error-focused evaluation to guide responsible deployment of LLM-based tools in peer review and other critical scientific workflows.

## 1 Introduction

The steady growth in both submissions and contributing authors to Artificial Intelligence (AI) conferences (e.g. ICML, ICLR, NeurIPS, etc) reflects the rapidly expanding global interest in AI research (Azad & Banu, 2024). In contrast to many traditional scientific disciplines where journals dominate as the primary publication channel (Vrettas & Sanderson, 2015; Kim, 2019), these conferences serve as leading platforms for disseminating state-of-the-art research and fostering community discussion and collaboration. At the current pace of innovation, conferences provide the most effective format for timely submission and visibility of new AI research, attaining influence and scholarly impact similar to many established first-tier journals (Kochetkov et al., 2021; Freyne et al., 2010; Keselman, 2019).

At the heart of these venues is the peer review system. The system has had to tolerate the strain of the increasing volume of submissions, contributing to a decline in review quality (Tropini et al., 2023; Horta & Jung, 2024; Kim et al., 2025; Chen et al., 2025). This is unsurprising, as producing a thoughtful and thorough review requires domain knowledge, time, and careful consideration—resources that may not scale with demand. Consequently, reviewers are increasingly turning to large language models (LLMs) as a means of managing their increasing workloads (Liang et al., 2024a; Latona et al., 2024). This shift is not simply confined to reviewers; authors have also begun leveraging LLMs within their research process to increase efficiency and sustain output amid intensifying competition (Bao et al., 2025; Liang et al., 2024b; Mishra et al., 2024).

In response, most conferences now permit the use of LLMs for writing assistance, provided that authors and reviewers explicitly disclose their use, and some have even started experimenting with incorporating AI into the review workflow[1]. Furthermore, recent work has begun to explore how LLMs can support the peer review process, and assess the extent to which they can perform this role effectively (Liu & Shah, 2023; Zhao et al., 2025; Yuan et al., 2022). However, much of this research has concentrated on *replacing* human reviewers with LLMs, leading to evaluation criteria that primarily measure the similarity between LLM-generated and human-generated review scores and comments.

In contrast, our work aims to *augment*, rather than replace, human reviewers. By shifting the focus from human–LLM similarity metrics to contradiction detection—a critical yet resource-intensive component of peer review—we reveal additional nuances in how LLMs can assist the review-writing process, beyond simply serving as substitutes for human reviewers. In summary, our contributions are as follows:

- **Contradiction Benchmark for Review Evaluation.** We introduce a benchmark derived from a diverse set of AI conference papers to systematically assess the ability of agentic review systems in detecting critical logical contradictions.

- **Knowledge-Graph–Based Contradiction Severity.** For fair and interpretable evaluation, we construct a paper-specific knowledge graph to quantify the severity and potential review impact of each identified contradiction, enabling a more nuanced assessment than binary correctness checks.

- **Multi-Layered Review (MLR) Framework.** We propose a novel agentic review system, whose key component lies in its "multi-layered" design: the system conducts multiple passes over the manuscript with progressively deeper levels of understanding before generating feedback. By emphasizing accurate comprehension of the manuscript and its claims, our framework supports a deeper analysis and more informative critiques, which is especially important for detecting subtle contradictions.

- **Thorough Evaluation of Proposed Framework.** We conduct a comprehensive evaluation of our proposed review system against the latest agentic review frameworks. Our analysis includes standard benchmarks comparing predicted review scores and textual similarity on ICLR, ICML, and NeurIPS submissions to demonstrate that our method is conference-agnostic. Beyond review comparisons, we assess system performance on our curated Contradiction Benchmark, a retraction dataset (WithdrarXiv-Check), a focus-level evaluation of facets emphasized or overlooked in reviewer feedback, and explicit manipulation in submissions. Collectively, these experiments provide a detailed characterization of the strengths and limitations of each system in both review accuracy and practical applicability.

- **Human Evaluations.** Based on feedback gathered from authors who used our system, it is perceived as beneficial within researchers' workflows. As is often the case with human peer review, authors expressed the most disagreement with the weaknesses identified by the agent, providing valuable feedback for future improvement.

## 1.1 Related Work

**Large Language Models for Peer Review**  Large Language Models (LLMs) have become integrated into many areas of research due to their incredible capabilities in reasoning (Kojima et al., 2022; Li et al., 2025; Mondorf & Plank, 2024), language comprehension (Liu et al., 2023; Guo et al., 2023; He et al., 2024), and content generation (Liu et al., 2024; Shakil et al., 2024; Luo et al., 2025; Kobak et al., 2025). Peer review is no exception. Indeed, a large-scale empirical study of review generation using GPT-4o showed that the overlaps between LLM and human feedback are comparable to those between two human reviewers (Liang et al., 2024c). In that work, an automated pipeline is proposed that first decomposes the paper into its

---

[1] https://iclr.cc/Conferences/2026/ReviewerGuide
https://neurips.cc/Conferences/2025/LLM
https://icml.cc/Conferences/2026/CallForPapers
https://aaai.org/conference/aaai/aaai-26/review-process-update/

constituent parts to construct a prompt for structured feedback. Notably, long texts are simply truncated to fit a smaller context window. In our subsequent sections, we refer to this system as LLM-Review.

Recent studies on review generation explore a multi-agent approach, where multiple instances of LLMs collaborate to improve review quality. These include works such as Multi-Agent Review Generation (MARG) (D'Arcy et al., 2024), AgentReview (Jin et al., 2024), the Automated Reviewer as part of the AI Scientist pipeline (Lu et al., 2024), and Generative Agent Reviewers (GAR) (Bougie & Watanabe, 2025). MARG consists of three different types of agents, a leader that delegates tasks and coordinates the other agents, worker agents, each receiving different chunks of the paper, and expert agents that specialize in specific sub-tasks. AgentReview uses three types of agents as well, reviewers, authors, and Area Chairs (AC) to exactly replicate the review process. A key feature of their approach is their design of various characteristics for each role such as a "knowledgeable" or "unknowledgeable" reviewer. As presented in the AI Scientist, the reviewer component, referred to as the AI Reviewer, generates reviews based on the NeurIPS template and employs self-reflection, few-shot examples, and response ensembling via meta-reviewing. In GAR, the paper is first parsed as a knowledge graph, then, as in AgentReview, 3-6 reviewers with different core attributes are randomly chosen to generate independent reviews before a meta-reviewer compiles them.

In addition, several frameworks have been introduced that train LLM agents on curated datasets. For example, ReviewAgents (Gao et al., 2025) develops a Review-CoT dataset that reformulates review comments and meta-reviews into structured chains-of-thought and includes references to relevant papers. Reviewer and area chair agents are then trained on them, and the usual peer review process is simulated to produce the final output. A related approach is Reviewer2 (Gao et al., 2024), where separate LLMs are fine-tuned for each step in a two-stage review system. In the first stage, the model generates aspect-specific prompts that highlight key areas for feedback, which are then used in the second stage to guide the final review generation.

Complementary to these approaches, our multi-agent system does not rely on ensembles of reviews, explicit reasoning techniques, or fine-tuning to enhance review quality. Instead, the main text and the appendix are processed separately by different agents, with an optional literature review conducted. The resulting information is then integrated by a review agent to generate the final review, with an initial focus on developing a deep understanding of the paper's content.

**Evaluation Methods for Generated Reviews**   Accurately assessing review quality requires evaluation across a relatively large number of papers. However, manual inspection at such scale is cumbersome and unrealistic. As a result, prior work has largely relied on qualitative measures for validation such as the correlation between LLM and human review scores, as well as similarity metrics like BLEU (Papineni et al., 2002), ROUGE (Lin & Hovy, 2003), and BertScore (Zhang et al., 2019) between review texts (Lu et al., 2024; Gao et al., 2024; 2025; Zhou et al., 2024). There are also works that use LLM as judges to evaluate how closely LLM-generated reviews align with human reviews (Liang et al., 2024c; Bougie & Watanabe, 2025; D'Arcy et al., 2024).

Given that peer review ultimately relies on human judgment, including human assessments of LLM-generated reviews is a natural step toward determining how well they meet reviewers' and author's expectations. Several works have collected such feedback on LLM-generated reviews but vary widely in the number of participants (Bougie & Watanabe, 2025; Robertson, 2023; Liang et al., 2024c; D'Arcy et al., 2024). Ranking-based evaluations offer an alternative means of assessing whether people prefer LLM-generated or human-written reviews. However, due to the scarcity of willing participants, these rankings are also often offloaded to LLM evaluators (Bougie & Watanabe, 2025; Tyser et al., 2024; Gao et al., 2025).

Independent from comparisons with human reviews, recent studies have started to evaluate the ability of LLMs for academic verification of research papers. Two of which are based on the WithdrarXiv dataset (Rao et al., 2024) that comprises of withdrawn papers from arXiv up to September 2024. Both concurrent works (Son et al., 2025; Zhang & Abernethy, 2025) filter the dataset using LLMs and manual inspections to ensure that each paper is identified with at least one clear error. However, they do not benchmark any review systems on their datasets, and instead evaluate LLMs using a prompt specifically designed to extract errors, thus only verifying the raw ability of the LLMs to find the errors. In a related work, to evaluate the robustness of GPT4 review generation, the authors applied two transformations separately to 20 NeurIPS

papers (Robertson, 2023). The first was to negate a key claim in the abstract and the second was to rewrite a random sentence to be informal. Adjacent to this, Skarlinski et al. (2024) introduces a benchmark for identifying contradictions within the scientific literature.

In contrast, our work designs an automated pipeline for error generation grounded in a knowledge graph extracted from each paper, ensuring precise control over the types of errors the system must detect, while also enabling scalability for reliable analysis. This paper-specific graph is then used to categorize each error by its severity.

## 2 Dataset and Benchmark

Effective error detection in peer review is essential for safeguarding the reliability and integrity of scientific findings. Because identifying substantive errors often requires careful cross-checking of assumptions, claims, and evidence across an entire manuscript, it is inherently resource-intensive. Leveraging LLMs to assist with this process therefore offers a promising avenue for alleviating reviewer burden. Motivated by this, we introduce a Contradiction Benchmark to assess whether review systems can integrate error detection directly into the review-generation process, enabling both error detection and review generation to be performed efficiently within a unified workflow.

### 2.1 Building the Contradiction Benchmark

To improve scalability and provide a clear definition of the errors that an agentic review system should detect, we propose an automated contradiction-generation pipeline to construct our Contradiction Benchmark, in which synthetic errors are systematically inserted into the original manuscripts. A key advantage of this workflow is its straightforward expansion to new papers over time, as well as its systematic control over the types and severity of contradictions. Furthermore, since the errors are artificially introduced, access to external information during review generation would not enable the system to directly find these errors through online searches. This helps ensure that successful contradiction detection reflects the capabilities of the system and the underlying LLM, rather than reliance on information retrieval. However, we caveat that a deliberately adversarial system could potentially identify the original manuscript on arXiv[2] and compare it against the edited version to recover the introduced contradictions. Consequently, while robust, the benchmark is not entirely resistant to adversarial exploitation. In Figure 1, we provide a schematic of our workflow, the core of which is to build a knowledge graph of the paper to be able to classify each error by its severity and potential impact on the paper's validity.

#### 2.1.1 Data Collection

We avoid over representing a small set of highly popular venues by including papers published at a broader range of leading AI conferences, thereby expanding both the scope and diversity of the benchmark. In total, we collected 257 papers published at ACL (Association for Computational Linguistics, 2025), AISTATS (International Conference on Artificial Intelligence and Statistics, 2025), CVPR (Computer Vision Foundation, 2025), and ICML (International Conference on Machine Learning, 2025) in 2025, as well as NeurIPS (Neural Information Processing Systems Foundation, 2024) in 2024, together with their LaTeX sources crawled from arXiv. We maintain a balanced representation of papers across each venue, with the exact paper counts summarized in Table 1. Additionally, to guarantee that we are legally allowed to modify and redistribute the content, our dataset is restricted to papers released under permissive Creative Commons licenses (CC BY, CC BY-SA, and CC0)[3].

#### 2.1.2 Knowledge Graph and Dataset Construction

**Motivation.** Initially, random sentences from the text were chosen and GPT-4.1 (Achiam et al., 2023) was tasked with rewriting the paragraph containing the sentence to form a contradiction. However, we noticed that without a structured approach for identifying where to introduce contradictions, the model sometimes

---

[2] https://arxiv.org/
[3] https://creativecommons.org/share-your-work/cclicenses/

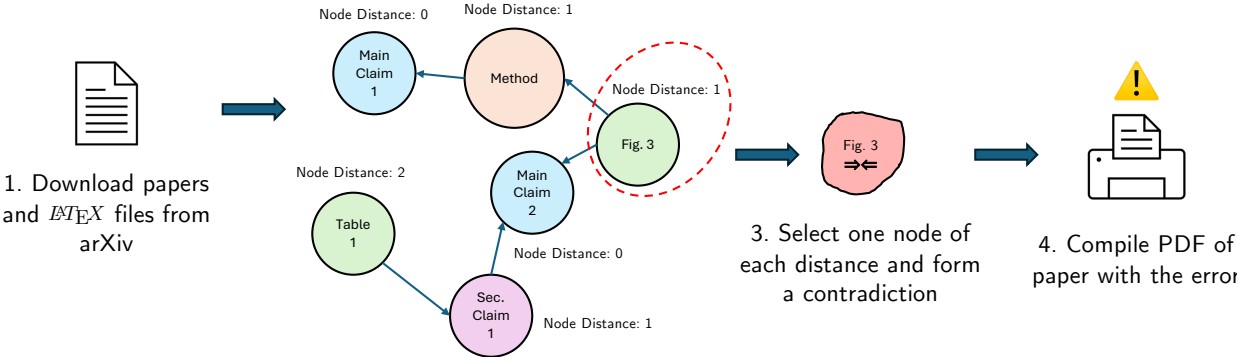

Figure 1: **Overview of Contradiction Benchmark Workflow.** 1) A random selection of papers and their LaTeX sources are downloaded from arXiv; 2) Using an LLM and specified graph structure, generate knowledge graph of paper; 3) Choose one node at each distance from a "main claim" node in the graph to form a contradiction; 4) Replace contradiction in LaTeX source and compile a PDF of the paper containing our constructed error.

Table 1: Metadata of papers used in the Contradiction Benchmark, categorized by conference venue and year. Included are the number of papers taken from each venue, the range of node distances in the knowledge graph, and the total number of contradictions successfully generated. Node distances are calculated as the number of edges from a "main claim" node while nodes with distance 0 *is* a "main claim" node. Each contradiction corresponds to a data point in our benchmark.

| Conference | Year | # Papers | Node Distance Range | # Contradictions |
| --- | --- | --- | --- | --- |
| ACL | 2025 | 51 | 0-8 | 242 |
| AISTATS | 2025 | 51 | 0-7 | 209 |
| CVPR | 2025 | 51 | 0-7 | 240 |
| ICML | 2025 | 51 | 0-7 | 238 |
| NeurIPS | 2024 | 53 | 0-7 | 235 |

targeted a claim in the abstract and, in other cases, minor implementation details—two types of errors that should not carry the same weight.

**Procedure.** To address this issue and systematically quantify the severity of different errors, we introduced an additional preprocessing step that constructs a knowledge graph of the paper prior to error generation, corresponding to Step 2 in Figure 1. We implement this using Gemini 2.5 Pro (Comanici et al., 2025), leveraging its constrained generation capabilities by providing a formal schema that defines the possible node types and their relationships within the graph. Then, we use the prompt template as in Figure 12 in the appendix for generation. Table 2 is a summary of the types of nodes and their definitions. When creating a node, we also require the LLM to clearly describe the point it represents and directly quote the exact text from which that point is derived. This facilitates error generation and subsequent LaTeX compilation of the contradiction-containing version of the paper.

For computing distances in the graph, we designate each "main claim" node as having a distance of 0, and assign all other nodes a distance equal to the length of the shortest path to any "main claim" node. These distances are defined to be inversely related to contradiction severity, such that contradictions associated

Table 2: Types of nodes defined for an LLM to generate a knowledge graph.

| Node Type | Subtype | Definition |
|---|---|---|
| Claim | Main
Secondary
Tertiary | High-level assertions that support the core contributions
Assertions that support main claims, but are not key results
Low-impact statements that supplement secondary claims |
| Evidence | High
Medium
Low | Strong results or empirical observations supporting a claim
Supporting findings that reinforce a claim but are not crucial
Minor observations with limited influence on conclusions |
| Methodology | - | A description or definition of a novel technique or approach |
| Implementation | - | Experimental details |
| Mathematical Theory | - | Newly proposed and proven mathematical theorems |

with more distant nodes are considered less severe. In Section 4.1, we observe that node distances are indeed inversely correlated with the proportion of contradictions detected by review systems, supporting the validity of our severity calibration.

Using the knowledge graph constructed for each paper, we create the final benchmark dataset by selecting nodes at varying graph distances to construct contradictions and compiling PDFs of the papers in which the respective content is replaced with the introduced contradictions (Steps 3 and 4 in Figure 1). Because each node distance corresponds to a distinct contradiction, the number of contradiction-containing variants of the original paper is equal to one more than its maximum node distance, including the "main claim" node at distance 0. This procedure yields approximately 200-250 data points per conference and a total of 1,164 overall as detailed in Table 1, with more details provided in Appendix A.1.

## 2.2 Automatic Evaluation

Due to the scale of the dataset, manual evaluation to determine the number of successfully detected contradictions is impractical. Consequently, we are motivated to propose an automatic evaluation of system performance on our benchmark. To this end, given that LLM evaluators are able to match the assessment quality of human evaluators (Chiang & Lee, 2023), we use the o3 reasoning model (OpenAI, 2025) and the prompt in Figure 17 in the appendix for evaluation. Given the review and the corresponding original and modified paragraphs, the LLM must predict whether the review identifies the contradiction correctly. We repeat the evaluation 10 times and report the average.

We selected o3 as the LLM-judge for two main reasons. First, when testing several models in a baseline evaluation on a subset of the unmodified papers (which contain no artificially introduced errors and therefore should always yield a "no" decision), o3 achieved a prediction accuracy of 99.9%. A more detailed discussion of this evaluation is presented in Appendix A.1. Second, upon manual inspection of its evaluation on the benchmark itself, we find that o3 had an acceptable and conservative performance as compared to GPT-4.1 and o4-mini.

## 2.3 Limitations

As contradictions are synthetically generated by an LLM and introduced into each paper, they may contain unnatural wording or structural patterns that make them easier to recognize than genuine errors. To determine the extent of this effect, we conduct a small audit on 50 contradictions using human raters. We ask each rater to evaluate whether the contradictions read naturally on a 5-point Likert scale from strongly disagree to strongly agree. Specifically, raters assess whether the text flows coherently from sentence to sentence and whether it avoids sounding obviously AI-generated. The full results are reported in Appendix B.1.

We find that 34% of contradictions are rated as lacking coherence across adjacent sentences, whereas only 8% are rated as sounding obviously AI-generated. The lower ratings for contextual flow are unsurprising,

Figure 2: **Schematic of our proposed Multi-Layered Review (MLR) System.** We use three different agents for each part of the process, an *Appendix Agent* (green), a *Literature Review Agent* (orange), and a *Review Agent* (pink). In the first step, the paper is chunked into its main text and appendix before the appendix is sent to the Appendix Agent for summarization. In the next step, the Literature Review Agent contextualizes the paper in the literature to position its contributions accurately. Then, the Review Agent performs a three-pass review that initially focuses on grasping the contents of the paper before synthesizing information from the other agents into a cohesive review.

as inserting a contradiction into an existing passage may inherently disrupt the contextual flow, even under human-written contradictions.

Although the contradictions in our benchmark may not always resemble genuine mistakes, this limitation should, in principle, make the benchmark easier. The relatively poor performance of the baseline review systems in Section 4.1 thus highlights a broader weakness: current systems struggle to identify errors even when unnatural textual patterns may be present. This observation is further supported by the results in Section 4.2, where all review systems, including MLR, perform worse on real mistakes than on our benchmark.

We therefore regard the Contradiction Benchmark as an initial step toward evaluating the error detection capabilities of LLM reviewers. A key direction for future work is to improve the benchmark by incorporating errors that better resemble naturally occurring mistakes.

# 3 Method

In this section, we describe our proposed Multi-Layered Review (MLR) system, which consists of three roles: an Appendix Agent, a Literature Review Agent, and a Review Agent, as summarized in Figure 2. All of our agents are powered by readily accessible, off-the-shelf LLMs. We begin by splitting the PDF of the paper into its main text (capped at 10 pages) and appendix. For all agents, we retain the original PDF format, as many LLMs are now able to accept files as input and analyze them effectively as images (Comanici et al., 2025; Anthropic, 2025; Achiam et al., 2023). This preserves key mathematical expressions and figures that are often difficult to accurately process into plain text. Due to limited context length of the models, we perform an in-depth analysis only on the main text. Moreover, during the development of the system, we observed that even when an LLM can process long inputs, its understanding degrades as input length increases.

**Appendix Agent.** Following the preprocessing (chunking) step, we send the appendix to a smaller model, Claude Haiku 3.5 (Anthropic, 2024), for a simple summarization. This component serves as a form of context management, reducing the likelihood that a weakness or question raised by the Review Agent has already been resolved in the appendix, which the Review Agent does not directly observe. The summarization focuses on extracting key experimental and implementation details, as these aspects are commonly discussed outside the main text and frequently factor into reviewer concerns.

**Literature Review Agent.** Our Literature Review Agent is powered by Claude Sonnet 4 (Anthropic, 2025) and contextualizes the contributions of the paper in the literature to assess their novelty and significance. We enable the use of a web search tool and give the agent autonomy to select the relevant sources; often this is sufficient to produce a comprehensive, elementary coverage of existing works. Including a literature review supplements the Review Agent's limited knowledge cutoff and enables it to integrate information beyond its training data, hence preventing mistaken weaknesses of "non-existent" material or novelty overestimation. However, for certain evaluations such as the WithdrarXiv-Check dataset where the LLM might find retraction comments online, we disable this component and consider this agent as optional within our workflow. The output of this agent is a literature review report that is shared with the Review Agent in its final step, together with the appendix summary.

**Review Agent.** The Review Agent (Claude Sonnet 4) is the primary driver of our MLR system and the main text is carefully analyzed by this agent. The novelty of our approach is to prioritize a deep *understanding* of its content prior to review generation, grounded in the natural process that human reviewers follow when evaluating a paper. To support this, we structure the agent's analysis into three progressive stages, drawing inspiration from the Three-Pass Approach for reading research papers (Keshav, 2007).

In the agent's first pass over the paper, we task it with producing a structured outline of the paper's core ideas at a high level. Then, using the outline as a guide, we prompt the agent to conduct a more detailed reading of the paper and enrich its notes with supporting evidence for key points. At this stage, the agent also examines the work critically, attending closely to potential weaknesses, assumptions, and gaps. The outcome of this phase is a coherent note reflecting the agent's thorough grasp of the paper's central claims and technical elements. Finally, the agent integrates the outputs of the Appendix and Literature Review Agents with selected points from its second pass insights to produce the final review, organized into the standard sections: Strengths, Weaknesses, Questions, Overall Recommendation, and Score, completing our three-pass prompt chain. We adopt the traditional NeurIPS 10-point scoring scale, recently replaced in 2025, and include its rubric in the prompt, while the other sections are given only as headers with no further instructions. More details and the exact prompts used can be found in Appendix A.2.

## 4 Results

Next, we present results on a comprehensive set of evaluations that provide insight into the strengths and limitations of our proposed Multi-Layered Review (MLR) system. We begin with error detection evaluations on our Contradiction Benchmark, as described in Section 2, and the WithdrarXiv-Check dataset (Zhang & Abernethy, 2025) (Sections 4.1-4.2). We then conduct a focus-level evaluation (Shin et al., 2025) to analyze the different facets that each review system focuses on relative to human reviewers in Section 4.3. Following that, we evaluate correlations between LLM-generated and human review scores, as well as their textual similarities, in Section 4.4 and examine the effects of explicit manipulation on LLM reviewers in Section 4.5. We conclude the section with a cost analysis (Section 4.6) and human evaluations (Section 4.7).

For comparison, we include LLM-Review (Liang et al., 2024c), the AI Reviewer (Lu et al., 2024), and AgentReview (Jin et al., 2024) as baselines. These selections are based on code availability, system diversity, and cost considerations. For each system evaluated, we largely follow the settings as specified in their original work. Additional details on the exact configurations used are provided in Appendix A.2.

### 4.1 Contradiction Benchmark

We follow the same evaluation method as outlined in Section 2.2 and report our results in Figure 3. For this benchmark, MLR uses only the Review Agent. As both AgentReview and the AI Reviewer use an ensemble of 3-4 reviews in their system, we likewise generate 4 reviews for fair comparison. We consider a contradiction to have been detected successfully if at least one of the reviewers identifies the error. For completeness, Appendix B.1, Figure 28 reports results for ensembles of 1–6 reviews and in Table 14, we provide the accuracies for all node distances 0-8 and the entire dataset. In the table, we compare our performance in both 4 review and single-review settings against the baselines. Notably, even with a single review, our system detects substantially more contradictions than other review methods, achieving an average accuracy improvement of more than 20%.

**Results.** Figure 3 is a plot of the accuracy of each review system in detecting contradictions, averaged over 10 runs using our LLM judge (o3) and grouped by node distance. We observe that **MLR achieves substantially higher contradiction detection accuracy than all baselines, with the largest gains on the most critical errors.** Specifically, MLR detects more than 70% of distance 0 contradictions—corresponding to a fourfold improvement over the next strongest baseline.

However, it should be noted that this improvement may be partially confounded by differences in each system's underlying LLM as all baselines use a variant of OpenAI's GPT, while MLR uses Claude models. To decouple the effect of model choice and system design, we perform an ablation by swapping the GPT model in LLM-Review to the same Claude model used in MLR. Its results are also shown in Figure 3 as LLM-Review+Claude. We find that changing the model alone leads to a considerable improvement in accuracy, particularly at distance 0, while MLR yields an additional large gain. Therefore, both factors contribute meaningfully to MLR's strong improvement. More detailed results and discussions can be found in Appendix B.1.

Across all systems, accuracy decreases with increasing node distance, indicating that our knowledge-graph-based severity classification is well calibrated. The exception at distance 6 is attributed to the limited sample size (18 papers), where each correct prediction shifts accuracy by approximately 5%.

The gray bars in Figure 3 show the number of papers per node distance. We restrict our analysis to distances 0-6, as only six samples exist beyond this range. Further, since distances 5 and 6 contain fewer than 50 papers, we report Wilson confidence intervals to provide robust uncertainty estimates under small sample sizes (Wallis, 2013).

### 4.2 WithdraXiv-Check Dataset

We further demonstrate the error detection capabilities of our MLR system (Review Agent only) through evaluation on the WithdraXiv-Check test dataset (Zhang & Abernethy, 2025) containing withdrawn papers from arXiv, along with their associated retraction comments. We follow the evaluation protocol outlined by the authors, using the o3 model, since the paper notes that the alternative judge, Gemini 2.5 Pro, is more lenient and sometimes hallucinates.

We use the prompt template, seen in Figure 22, to determine if there is an exact match in the review to the retraction comment. However, upon careful inspection of the results, we find instances where a review system correctly identifies a problematic aspect of the paper, but does not exactly match the corresponding retraction comment, which is itself somewhat vague. We illustrate this with one such example in Figure 4, where the review has correctly mentioned a weakness in Lemma 9, similar to the retraction comment, yet the LLM-judge does not consider this to be an exact match. Therefore, we introduce a more relaxed setting whereby an exact match is not required, but if both the review and retraction comment mention a similar problem, it is considered a successful detection. To distinguish these settings, in Table 3, we refer to the former, original prompt as "Exact" and the latter, modified version as "Similar". The changes of the prompt in the "Similar" setting can be found in Appendix A.3.

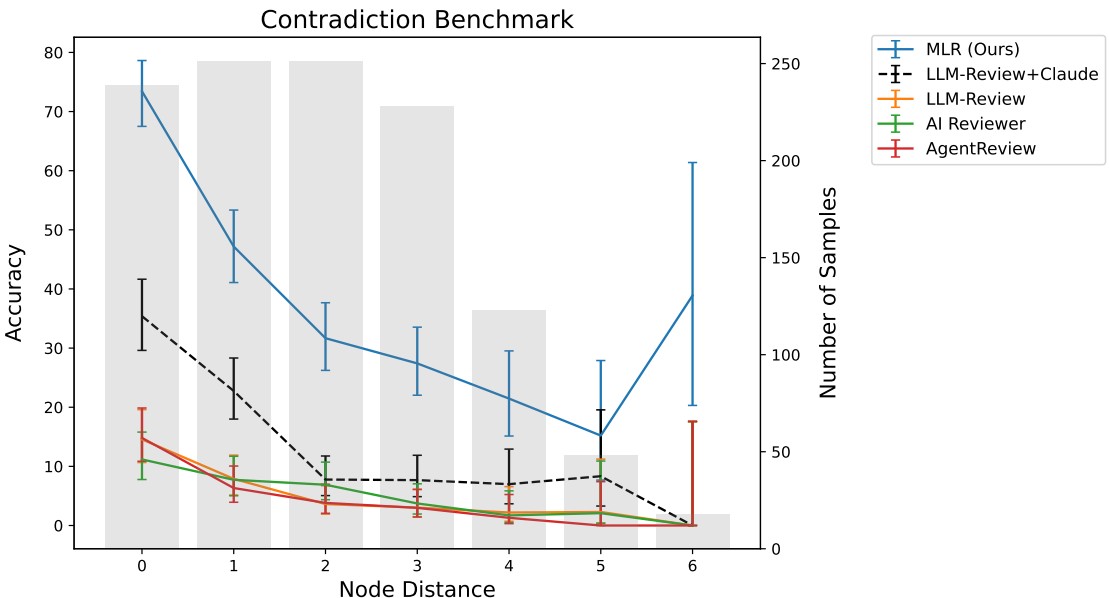

Figure 3: Line plot of the accuracies of various review systems in successfully catching errors in our Contradiction Benchmark by node distance. Included are bar plots in gray that show the number of data points in the benchmark for each node distance, as well as an ablation on model choice using LLM-Review with Claude Sonnet 4 and no truncation. As accuracy can be misleading with small sample sizes, we plot Wilson confidence intervals alongside the accuracy curves.

**Retraction comment**

> Due to a flaw in Lemma 9, the paper has been withdrawn

**Weakness in review**

> **Missing Rigorous Proof of Bridge-Freeness Preservation**: While Theorem 4 claims that G has a C-augmenting set A such that G - A is bridge-free, the proof relies on Lemmas 7 and 9 whose correctness is not convincingly established, particularly the intricate algorithm in Lemma 9.

Figure 4: Example of a vague retraction comment and a weakness raised by our MLR system that was *not* considered as an exact match by the LLM-judge.

**Results.** In Table 3 we find that **our MLR system outperforms all baselines for both settings by a considerable margin**, exceeding the next best system by at least 5%. We also observe that, as compared to the Contradiction Benchmark in the previous section, the improvement is less pronounced. Possible explanations include that errors in the dataset are generally easier to detect due to their synthetic generation (Section 2.3), and that the review systems evaluated are optimized for machine learning conferences, whereas the papers in WithdrarXiv-Check predominantly come from theoretical mathematics and physics journals. A more detailed discussion can be found in Appendix B.2.

### 4.3 Focus Distribution

In this section, we conduct a focus-level evaluation, following the framework proposed in (Shin et al., 2025), to identify the facets emphasized by each review system. On this evaluation, MLR uses both the Appendix and Review agent with a web search in place of a full literature review. As the code for their automatic

Table 3: Accuracy of detecting errors in the WithdrarXiv-Check dataset for two different settings in the LLM-judge. "Similar" captures cases where the review raises concerns comparable to the retraction comment, whereas "Exact" refers to exact matches. **Bolded** values indicate the highest accuracy.

| Setting / Method | MLR (Ours) | LLM-Review | AI Reviewer | AgentReview |
|---|---|---|---|---|
| Similar | **26.07** | 5.21 | 13.74 | 18.48 |
| Exact | **16.11** | 2.37 | 9.00 | 5.69 |

Table 4: Target and aspect facets used to categorize each strength and weakness to reflect its specific focus.

| Target | Aspect |
|---|---|
| Overall Motivation | Communication Clarity |
| Method | Validity |
| Theory | Novelty |
| Experiment | Impact |
| Conclusion | Not-specific |
| Paper | |
| Prior Research | |

evaluation pipeline was not released, we implement their method based on the prompts and the provided dataset. More details can be found in Appendix A.4. Briefly, each system or model generates reviews of the papers in their Expert Review Dataset, then strengths and weaknesses are extracted from all the reviews. An LLM annotator (o3-mini) labels each strength and weakness with a target and an aspect facet that corresponds to their focus. Targets capture what a strength or weakness comments on, while aspects describe the particular properties of that target under evaluation. Table 4 lists the facets introduced in their work that we adopt in our analysis. The Prior Research facet applies only to weaknesses, whereas Paper and Not-specific refer to general comments that do not correspond to any particular target or aspect.

Calculating the proportion of strengths and weaknesses that correspond to each target and aspect facet results in four distributions that the authors name focus distributions. To investigate the behavior of LLMs and review systems when reviewing papers, their framework compares the focus distributions of the automated systems and humans using the Kullback–Leibler (KL) divergence. We follow suit and substantially expand the set of models used in their evaluation to include the latest frontier LLMs (OpenAI, 2025; DeepMind, 2025). When evaluating the models themselves, we use the review prompt exactly as presented in their paper. For the human and GPT-4o mini focus distributions, we use their released data.

**Results.** Table 5 reports the KL divergences and shows that model-generated reviews with simple prompts align more closely with human focus in strengths, but diverge more in weaknesses compared to review systems. One possible explanation is that identifying strengths is generally easier, as authors often explicitly state and emphasize them in manuscripts, for example through summaries of their contributions. As a result, simple prompting may be sufficient to extract such information, whereas agentic review systems are typically designed to adopt a more critical stance and are less likely to accept authors' claims at face value. In contrast, identifying weaknesses typically requires deeper reasoning and verification, which may explain why agentic systems align more closely with human focus distributions in this setting.

Delving deeper into MLR's focus distributions as well as those that differ the most with human reviewers, we present radar plots of each target and aspect distribution for strengths in Figure 5 and weaknesses in Figure 6. In the strength's target plot, we include the AgentReview's distributions and observe that, while human reviewers have a tendency to provide more general comments on the manuscript (Paper facet), both our MLR system and the AgentReview system exhibit a lower proportion of such comments and instead adopt more specific focuses. Another notable difference is the greater emphasis placed on the experimental facet by review systems compared to humans, suggesting that automated reviewers tend to ground their feedback more heavily in reported empirical results, which are easier to verify. These are corroborated in

Table 5: KL divergence between the focus distributions of each model/review system and human reviewers. A lower value indicates that the model/system has a focus distribution that is more similar to that of human reviewers. ST represents the focus distribution over target facets and SA represents the focus distribution over aspect facets within the review's strengths; analogously, WT and WA represent the corresponding distributions for weaknesses. S-Avg is the average of the KL divergences for the strength distributions, similarly for W-Avg with respect to the weaknesses, and the last column is the average over the KL divergences of all four distributions. * denotes results obtained using data from prior work; highlighted in green and red are the lowest and highest values in the column respectively.

| Model/System | ST | SA | S-Avg | WT | WA | W-Avg | Avg |
|---|---|---|---|---|---|---|---|
| GPT-4o mini* | 0.083 | 0.047 | 0.065 | 0.054 | 0.441 | 0.248 | 0.156 |
| GPT-5 mini | 0.202 | 0.109 | 0.156 | 0.227 | 0.190 | 0.209 | 0.182 |
| GPT-5.1 | 0.149 | 0.058 | 0.104 | 0.108 | 0.491 | 0.300 | 0.202 |
| Gemini 2.5 Pro | 0.052 | 0.069 | 0.061 | 0.142 | 0.171 | 0.157 | 0.108 |
| Gemini 3 Pro | 0.207 | 0.121 | 0.164 | 0.116 | 0.091 | 0.104 | 0.134 |
| Claude Sonnet 4.5 | 0.142 | 0.107 | 0.125 | 0.047 | 0.090 | 0.069 | 0.097 |
| MLR (Ours) | 0.309 | 0.322 | 0.316 | 0.143 | 0.184 | 0.164 | 0.240 |
| LLM-Review | 0.161 | 0.182 | 0.172 | 0.087 | 0.056 | 0.072 | 0.121 |
| AI Reviewer | 0.267 | 0.152 | 0.210 | 0.180 | 0.432 | 0.306 | 0.266 |
| AgentReview | 0.311 | 0.286 | 0.299 | 0.028 | 0.087 | 0.058 | 0.178 |

the aspect distribution, where validity is disproportionally high for MLR and AgentReview. In contrast, humans tend to focus on communication clarity in the paper.

Regarding weaknesses, we plot those of our MLR system and GPT-5 mini for the target distribution and GPT-5.1 for the aspect distribution. Consistent with the strengths analysis, we see a lower proportion of general (Paper facet) comments from our MLR system, whereas GPT-5 mini shows a proportion comparable to that of human reviewers. Furthermore, human reviewers tend to emphasize communication clarity and novelty when identifying weaknesses, while our MLR system places greater emphasis on validity. GPT-5.1 appears to follow a similar trend, prioritizing validity over novelty. These findings support the high error detection rate of our MLR system in Section 4.1 as well as a strong specificity rating in Section 4.7.

Taken together, our results indicate that **the MLR system offers an alternative review perspective that complements human reviewers, contributing to a more comprehensive peer-review process**. For radar plots of the other models and review systems evaluated, refer to Appendix B.3.

### 4.4 Evaluation on Conference Submissions

From the focus-level evaluation, we observed that our MLR system generates reviews that focus on substantially different facets in both the strengths and weaknesses. While this is advantageous in providing new perspectives and diversifying reviews, thereby helping uncover additional reasons to accept or reject the paper, it becomes problematic if the system's final recommendation diverges entirely from human judgment.

For this reason, we are motivated to evaluate the semantic similarities and score correlations of our proposed system on a range of conference submissions. In contrast to prior studies that concentrate on ICLR papers, likely due to the full public accessibility of all submissions regardless of decision outcome, we crawl an even split of accepted and rejected papers from ICLR, ICML, and NeurIPS, where the rejected papers are those that opted in for public release. Table 6 summarizes the papers used in our analysis, with further experimental details and results provided in Appendix A.5 and B.4 respectively. Similar to the focus-level evaluation, in this section, MLR uses both the Appendix and Review agent with a web search in place of a full literature review.

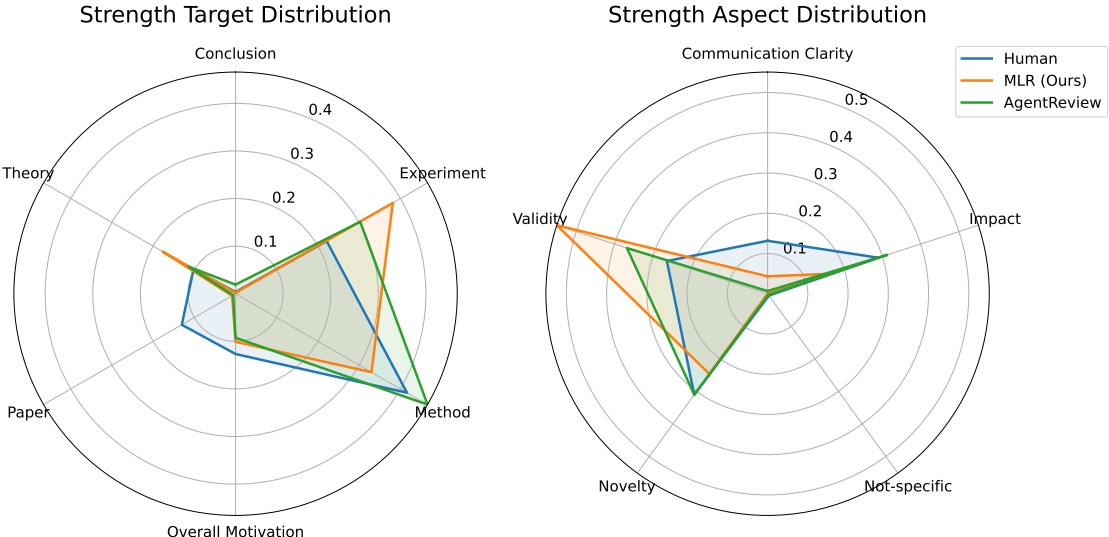

Figure 5: Radar plot of the strength target and strength aspect focus distributions for human reviews, our MLR system reviews, and AgentReview reviews.

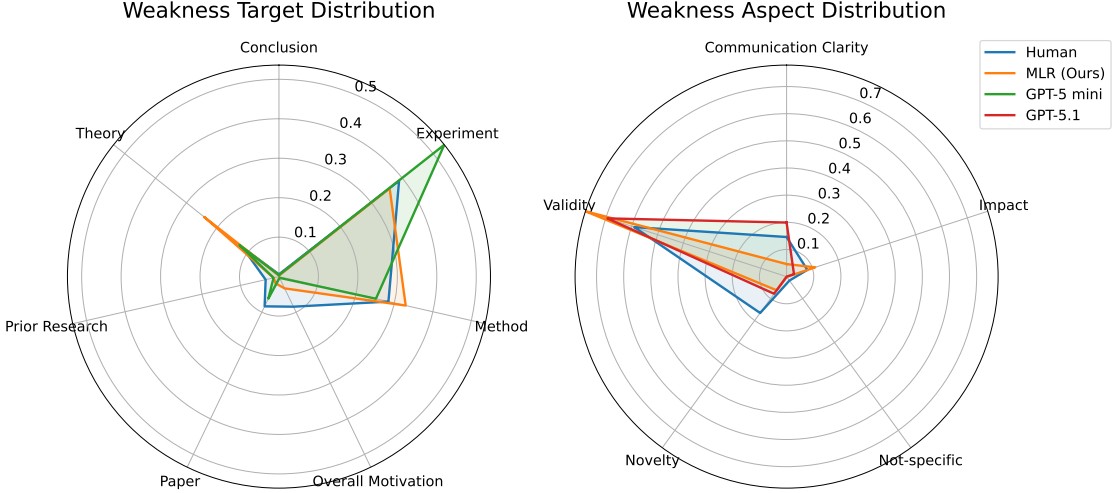

Figure 6: Radar plot of the weakness target and weakness aspect focus distributions for human reviews and our MLR system reviews. For the target plot, we include GPT-5 mini's distribution and for the aspect plot, we include GPT-5.1's distribution as well.

### 4.4.1 Semantic Similarity

**Evaluation metrics.**  To assess semantic consistency between LLM-generated and human written reviews, we use metrics BLEU-4 (Papineni et al., 2002), ROUGE-1, ROUGE-L (Lin & Hovy, 2003), and BertScore with RoBERTa-large (Zhang et al., 2019). Since each paper typically has multiple human reviews, we evaluate each candidate review against all available human reviews for that paper as a multi-reference set. BLEU-4 measures the quality of the generated text by evaluating how many 1 to 4-gram sequences it shares with the human review. ROUGE-1 evaluates the overlap of individual words between the generated and reference texts and ROUGE-L does the same with the longest common subsequences. Whereas, rather than relying on string-matching, BertScore compares texts by computing the cosine similarity between their token embeddings. For readability, we multiply each score by 100 and therefore BLEU-4, ROUGE-1, and ROUGE-

Table 6: Metadata of papers used in the evaluation on conference submissions. We list the number of papers by conference, year, and decision.

| Conference | Year | Decision | # Papers |
|---|---|---|---|
| ICML | 2025 | Accept | 51 |
| | | Reject | 50 |
| ICLR | 2025 | Accept | 50 |
| | | Reject | 50 |
| NeurIPS | 2024 | Accept | 52 |
| | | Reject | 50 |

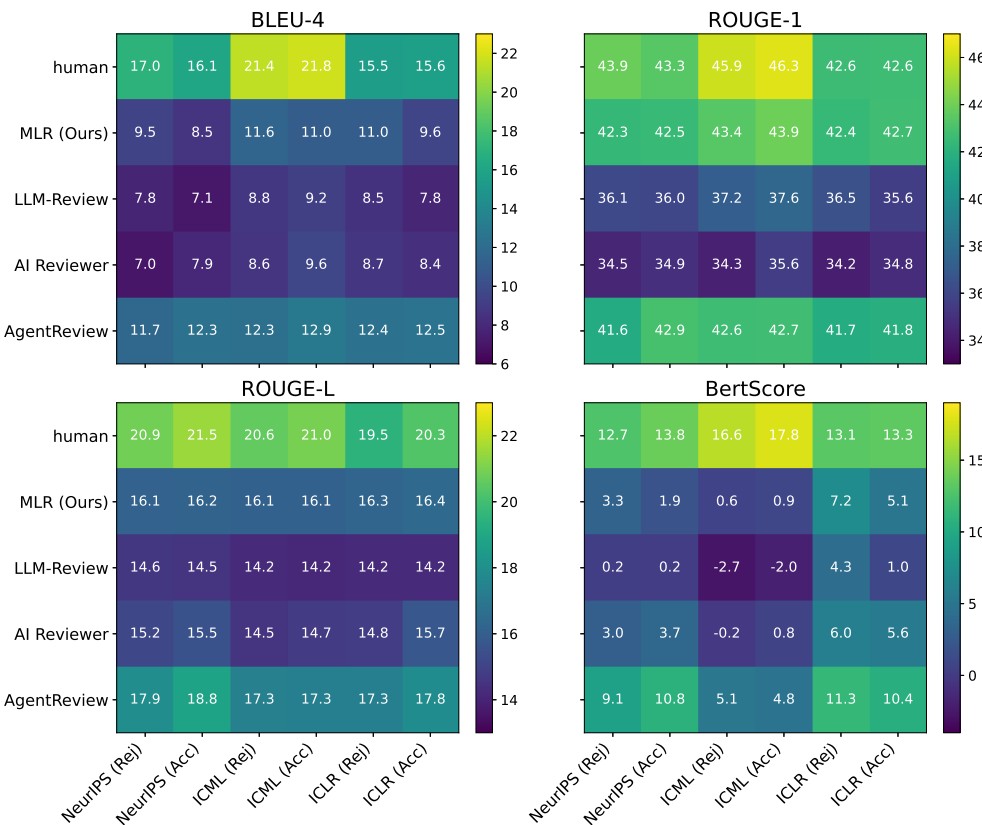

Figure 7: BLEU-4, ROUGE-1, ROUGE-L, and BertScore between each review system and human reviews, as well as within the human reviews themselves. We report results separately by conference and decision outcome (shown on the x-axis), while the y-axis lists the entity that generated the reviews. "Acc" indicates accepted papers while "Rej" indicates rejected papers. For all metrics, values closer to 100 indicate greater similarity.

L can have values between 0 to 100, while BertScores range from -100 to 100. A higher score corresponds to a stronger degree of semantic similarity.

**Results.** We observe that **MLR reviews are semantically different from human reviews.** Figure 7 presents a heat map of the semantic consistency between each review system and human reviews. For reference, we also include an evaluation of all metrics within the human reviews themselves, these are featured in the top row of each subplot. Generally, our findings corroborate with those in the focus-level evaluation (Section 4.3) since our MLR system has relatively low scores across most metrics, except for ROUGE-1, and

has the largest divergence in focus facets from human reviews. A reason for the high ROUGE-1 could be that while the reviews generated by our MLR system emphasizes different areas of strengths and weaknesses, they still contain similar vocabulary involving technical phrases to the human reviews.

Interestingly, even among human reviews themselves, BertScore is relatively low at about 12-18 across conferences, suggesting **substantial diversity in how reviewers interpret and respond to a paper as well**. This provides a compelling basis for integrating LLMs with distinct perspectives into the review process, as increased diversity in judgments often leads to more comprehensive and balanced evaluations. On the other hand, AgentReview has the strongest semantic consistency out of all the evaluated review systems, which aligns with its objective of realistically simulating the full peer review process to study the roles and influences of different participants involved.

### 4.4.2 Correlation

**Evaluation metrics.** Next, we consider the correlation between agentic system review scores and human review scores. To quantify this, we use the Pearson (Benesty et al., 2009), Spearman (Spearman, 1904), and Kendall's Tau (Kendall, 1938). Pearson correlation reflects how strongly two variables are linearly related, whereas Spearman and Kendall correlations capture rank-based relationships that are potentially non-linear. Since LLM-Review does not output any scores in their review, we turn to GPT-4.1 to predict the score of a paper from the non-numeric sections of reviews generated by each system using the prompt in Figure 23. For systems with more than one review per paper, we take the average score of the set of reviews. These results can be found in Table 7. As a reference, we also include the correlations between score predictions from human reviews and their true scores, which are highlighted in gray in the table.

**Results.** We start by evaluating GPT-4.1's ability to predict a paper's score from its reviews to ensure that our subsequent analysis based on these predicted scores is meaningful. In Figure 8, we plot the true and predicted scores of human reviews, where each subplot corresponds to the results of the different conferences. Green and red dots indicate accepted and rejected papers respectively, while the gray diagonal line is a reference for perfect predictions. As seen in the figure, and corroborated by the gray-shaded rows of Table 7, the predictions of GPT-4.1 align closely with human assessments, yielding Pearson correlations approaching 0.7 to 0.8 across conferences. Therefore, the predicted scores provide a reliable basis for further analysis.

From Table 7, we observe that, in all conferences, our MLR system consistently achieves relatively high correlations between its predicted scores and the true scores. We also include the correlation between the scores generated by each system, if they do, and true human review scores in the appendix, Table 18. Those evaluations further validate our findings here. **Overall, these results demonstrate that our system is venue-agnostic within the main machine learning conferences and, while providing new perspectives to the current peer review process, importantly, remains well aligned with human judgments of submission quality.**

Lastly, recognizing the importance of distinguishing between papers that meet the standard for acceptance and those that do not, we include a box plot of the LLM-predicted scores for each system and the true human review scores in Figure 9, grouped by decision outcome and venue. To minimize visual clutter, we display only the whiskers. Red and green dots represent the mean scores within rejected and accepted papers respectively, while the long lines correspond to the medians. As seen in the figure, while there is a large overlap between the box plots of accepted and rejected papers in the human review scores for all venues, there is still a clear distinction in their average and median scores. This is consistent with the role of review scores as a key determinant of paper acceptance outcomes.

In comparison, LLM-Review and AgentReviewer show substantial overlap in all of their green and red box plots, with nearly identical mean scores for each decision outcome, suggesting that they are unable to differentiate between accepted and rejected papers reliably. Meanwhile, although the red box plots subsume the green ones for both our MLR system and the AI Reviewer, their corresponding mean values remain clearly separated across all conferences, with a larger gap observed for MLR. These imply that **the systems have more difficulty and variance in scoring rejected papers, but are capable of identifying quality submissions worthy of acceptance.** A possible explanation is that NeurIPS and ICML release

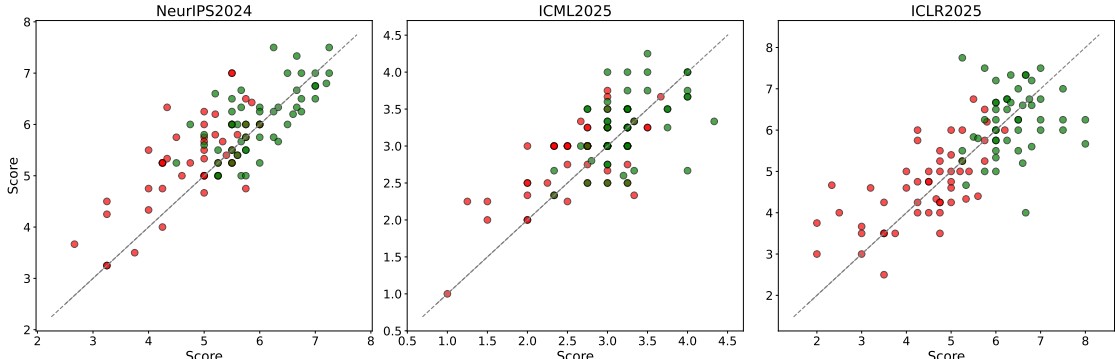

Figure 8: Plot of the predicted scores by GPT-4.1 against the true scores of human reviews, split by venue. A perfect prediction will lie on the gray, diagonal line ($y = x$). Green and red dots represent accepted and rejected papers respectively.

Table 7: Pearson, Spearman, and Kendall's Tau correlation between the predicted scores of model-generated and human reviews across venues. Highest values, excluding the human baseline, are **bolded** while second highest values are underlined.

| Venue | Method | Pearson | Spearman | Kendall |
|---|---|---|---|---|
| | Human (Reference) | 0.781 | 0.700 | 0.547 |
| | MLR (Ours) | **0.451** | 0.386 | 0.299 |
| NeurIPS 2024 | LLM-Review | 0.358 | **0.457** | **0.377** |
| | AI Reviewer | 0.328 | 0.331 | 0.265 |
| | AgentReview | 0.167 | 0.139 | 0.103 |
| | Human (Reference) | 0.684 | 0.599 | 0.477 |
| | MLR (Ours) | 0.429 | 0.333 | 0.277 |
| ICML 2025 | LLM-Review | 0.169 | 0.081 | 0.070 |
| | AI Reviewer | **0.439** | **0.416** | **0.353** |
| | AgentReview | 0.006 | 0.054 | 0.049 |
| | Human (Reference) | 0.742 | 0.754 | 0.577 |
| | MLR (Ours) | **0.586** | **0.574** | **0.472** |
| ICLR 2025 | LLM-Review | -0.013 | -0.006 | -0.003 |
| | AI Reviewer | 0.538 | 0.453 | 0.369 |
| | AgentReview | 0.195 | 0.204 | 0.172 |

only opted-in rejected reviews, and authors tend to opt in when their papers are relatively strong and close to acceptance. It is also worth noting that our MLR system exhibits both the largest range and lowest scores for rejected ICLR papers—where all reviews, including those for the weakest submissions, are released—indicating that our system can distinguish these submissions more effectively.

### 4.5 Explicit Manipulation

To evaluate the robustness of LLM reviewers, we tested each system on papers containing explicit manipulations, unseen by human reviewers. In this experiment, MLR uses only the Review agent. Following the set-up in Ye et al. (2024), we inject text designed to influence the judgments of LLM reviewers after the conclusion section of the paper (see Appendix A.6 for details). In our experiment, we crawled 50 rejected papers from arXiv, most of which were originally submitted to ICLR 2025. Presented in Table 8 are both the

---

[4]Plots for MLR, LLM-Review, and AI Reviewer are missing whiskers as they only output a single review or a meta-review for each paper. Therefore, scores can only take specific integer values resulting in many papers receiving the same scores. On the other hand, AgentReview and Human have at least three reviews per paper and we take their average for these plots, hence they have a larger range of values.

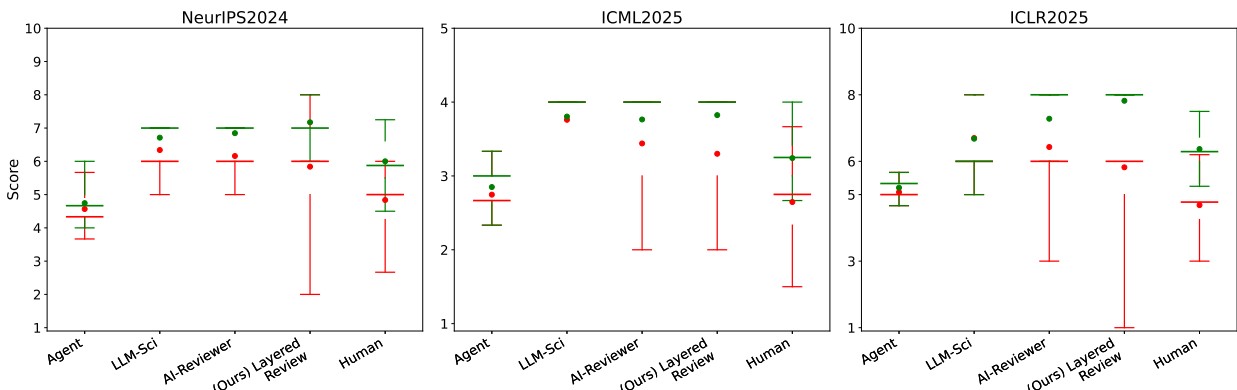

Figure 9: Box plot of true human review scores and each system's LLM-predicted scores, categorized by accepted (green) and rejected (red) papers for each venue. The long bars represent the median while the dots represent the mean scores. Short lines at the extremes of each plot are the whiskers denoting the furthest data point lying within 1.5 times of the first and third quartile[4].

Table 8: Mean and standard deviation of LLM-judge predicted scores and each review system's overall scores (Self) on rejected conference papers before and after explicitly manipulating them. $\Delta$ indicates the mean and standard deviation of the change in scores, **bolded** are the smallest change, and underlined are the second smallest.

| Method | LLM-judge | | | Self | | |
|---|---|---|---|---|---|---|
| | Before | After | $\Delta$ | Before | After | $\Delta$ |
| MLR (Ours) | $7.30_{\pm1.10}$ | $8.00_{\pm2.88}$ | $\mathbf{0.70}_{\pm2.46}$ | $6.44_{\pm0.67}$ | $6.70_{\pm2.10}$ | $\mathbf{0.26}_{\pm1.85}$ |
| LLM-Review | $6.88_{\pm1.32}$ | $9.30_{\pm1.47}$ | $2.42_{\pm1.47}$ | - | - | - |
| AI Reviewer | $7.16_{\pm1.05}$ | $7.88_{\pm0.62}$ | $\underline{0.72}_{\pm0.98}$ | $6.68_{\pm1.16}$ | $8.00_{\pm0.66}$ | $1.32_{\pm1.09}$ |
| AgentReview | $5.51_{\pm0.69}$ | $7.19_{\pm0.98}$ | $1.68_{\pm1.09}$ | $5.68_{\pm0.36}$ | $6.11_{\pm0.56}$ | $\underline{0.43}_{\pm0.69}$ |

scores predicted by the LLM-judge, as in the previous section, as well as the impact of these modifications on scores produced by the review systems themselves. The latter scores are referred to as "Self" in the table.

**Results.** While our MLR system has the smallest mean score change in Table 8, we notice a high standard deviation for changes in both LLM-judge and Self scores, at 2.46 and 1.85 respectively. Upon further investigation into the reviews themselves, we find that all systems are strongly influenced by the injected text, corroborating the large score changes and standard deviations in the table. However, in 8 out of 50 cases, MLR detected the presence of such manipulative texts, most of which are treated as severe ethical violations, while others are considered as "accidents" to be removed, suggesting comparatively greater robustness. We attribute this detection capability to two factors: i) MLR system's design, which prioritizes deep comprehension of the paper over blind reviewer instructions, and ii) the multimodal input to Claude where the PDF is provided in both image and text form, as the injected content is not visible in the image itself. **These factors suggest both the susceptibility of LLM review systems to adversarial manipulation and potential safeguards against such unethical practices in the future**.

We further observe that the AI Reviewer and AgentReview are only mildly affected when scores are produced by the LLM-judge and the review system themselves, respectively. A possible reason for the AI Reviewer's small score change is its use of a meta-reviewer that does not receive the injected instruction, together with an additional limitations section in their NeurIPS review template. This section is not targeted by the injected content, allowing more reliable scoring by the LLM judge. Conversely, the minimal score change observed for AgentReview is likely an artifact of the system's scoring procedure, as their review scores generally exhibit low variance and fall between 4 and 6, as seen in Appendix B.4.

Table 9: Token count, unit price, and total cost of each LLM review system. All prices are in USD and we consider input and output tokens separately. As the literature review agent in our MLR system is optional, usually not used for analyses, and other review systems do not incorporate web searches, we leave that out of our total for fair comparison. It is included in the table as a shaded gray row for reference with "opt." indicating that this is an optional component.

| Method | Token count | | Unit price ($/1M tokens) | | Cost ($) | |
|---|---|---|---|---|---|---|
| | Input | Output | Input | Output | Input | Output |
| MLR (Ours) | 189,062 | 3,913 | 3 | 15 | 0.42 | 0.05 |
| Appendix | 66,521 | 1,011 | 0.8 | 4 | 0.05 | < 0.01 |
| Review | 122,541 | 2,902 | 3 | 15 | 0.37 | 0.04 |
| Literature Review (opt.) | 312,767 | 2,032 | 3 | 15 | 0.94 | 0.03 |
| LLM-Review | 6,517 | 740 | 2 | 8 | 0.01 | < 0.01 |
| AI Reviewer | 403,654 | 12,448 | 1.1 | 4.4 | 0.44 | 0.05 |
| AgentReview | 310,964 | 3,404 | 2.5 | 10 | 0.78 | 0.03 |

### 4.6 Cost Analysis

In Table 9, we compare the token count and cost of each agentic review system, split by input and output tokens, as the unit costs are usually substantially higher for outputs. To ensure fair comparison, we exclude the optional literature review agent in our MLR system since other LLM reviewers do not employ any web searches, and we exclude a literature review during evaluations where access to the internet would confound the results.

As observed in the table, LLM-Review incurs the lowest cost due to its simple system design and strict truncation of the paper, resulting in a low input token count. However, for the same reason, their reviews are unable to reliably detect errors in the paper (Figure 3, Table 3) nor differentiate between high- and low-quality papers (Figure 9).

The next lowest costs belong to our MLR system and the AI Reviewer, both of which incur about USD 0.50 per review. Notably, MLR is approximately twice as efficient as the AI Reviewer in terms of token usage. The comparable overall costs arise primarily from our use of an LLM with higher unit prices—a controllable design choice, as the underlying model can be easily swapped to reduce costs—whereas reducing token usage would require a more substantial system redesign. Furthermore, in Appendix B.4, we consider a more token-efficient variant of our system by consolidating the three-prompt chain into a single prompt and show that it achieves comparable performance while reducing the overall cost by approximately two-thirds.

### 4.7 Human Evaluation and Feedback

To complement our quantitative benchmarks, we conducted a qualitative user study involving $N = 38$ distinct review sessions. The participants consisted of active researchers and authors who uploaded their manuscripts to the full MLR system including all agents. To improve their user experience, we introduced an additional "To-do List" section in our reviews, providing actionable recommendations to strengthen a paper. For each session, we collected high-level feedback on five global metrics—Helpfulness, Reuse Intention, Perceived Benefit (Beneficiality), Criticality, and Manuscript Improvement (Improvement)—measured on a 5-point Likert Scale. Additionally, we gathered granular feedback on each review section to assess the *Accuracy* and *Specificity* of the generated content. Participants also retained the option to indicate if they agreed or disagreed with any specific review comment using a simple thumbs-up or thumbs-down response. In total, the study yielded 262 section-specific ratings and 725 distinct feedback on review comments. Details regarding the questionnaire design and rubric definitions are provided in Appendix A.7.

**Global Satisfaction Metrics.** We computed the Mean Opinion Scores (MOS) for the five global metrics using the feedback for the human study. The MOS is computed as the average feedback scores on the 5-point

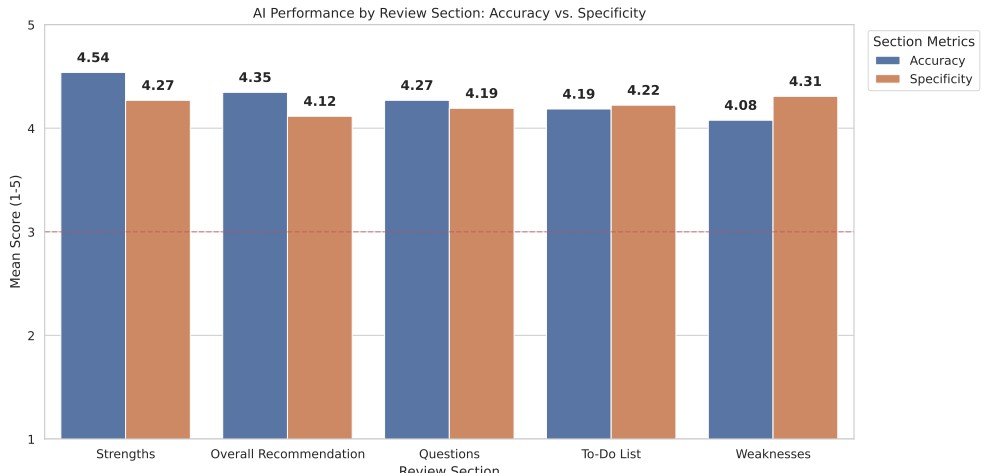

Figure 10: Distribution of Accuracy vs. Specificity scores for distinct review sections. The red horizontal line at a score of 3 indicates the neutral rating.

Table 10: Granular Analysis of Review Comments Feedback. For each review section, we report the total number of comments interacted by participants (*Reported Comments*), along with how many were judged correct (*Agreed Comments*). Parentheses show the corresponding acceptance rates (Comment Acceptance Rate, CAR = agreed / reported).

| Section | Reported Comments | Agreed Comments (Comment Acceptance Rate) |
|---|---|---|
| Overall Recommendation | 31 | 29 (94%) |
| Strengths | 90 | 83 (92%) |
| Questions | 68 | 54 (79%) |
| To-Do List | 107 | 83 (78%) |
| Weaknesses | 82 | 56 (68%) |
| Total | 378 | 305 (81%) |

scale. Our system achieved its highest ratings for *Beneficiality* ($\mu = 4.25$) and *Improvement* ($\mu = 4.04$), indicating that users found the feedback produced highly constructive and actionable. *Helpfulness* ($\mu = 3.70$) and *Reuse Intention* ($\mu = 3.74$) were also rated favorably, suggesting a strong willingness to integrate the system into their research workflow. Notably, *Criticality* received a score closest to the neutral baseline ($\mu = 2.88$). In our scoring rubric, a score of 3 for Criticality denotes a "balanced" tone. This implies that the system avoids being overly agreeable while maintaining an analytical, rather than adversarial, persona—a desirable trait for an assistive drafting tool.

**Section-wise Accuracy and Specificity.** We decompose the reviews into their constituent sections (Strengths, Weaknesses, Questions, Overall Recommendation, To-Do List) to analyze variation in performance across sections. As shown in Figure 10, all sections consistently achieved high accuracy and specificity scores, at least one point above a neutral rating. However, the *Weaknesses* section was rated the least accurate by the users. This observation is supported by our sentiment acceptance analysis (Table 10), where the acceptance rate is computed as the fraction of comments in each section that users agreed with. Comments in the Weaknesses section had the lowest acceptance rate among all categories (68%). In contrast, the To-Do List achieved a high acceptance rate (78%). This discrepancy implies that while the system is adept at synthesizing actionable next steps and summarizing contributions, its assessments are occasionally perceived as misplaced by authors and was the least well-received among the other comments.

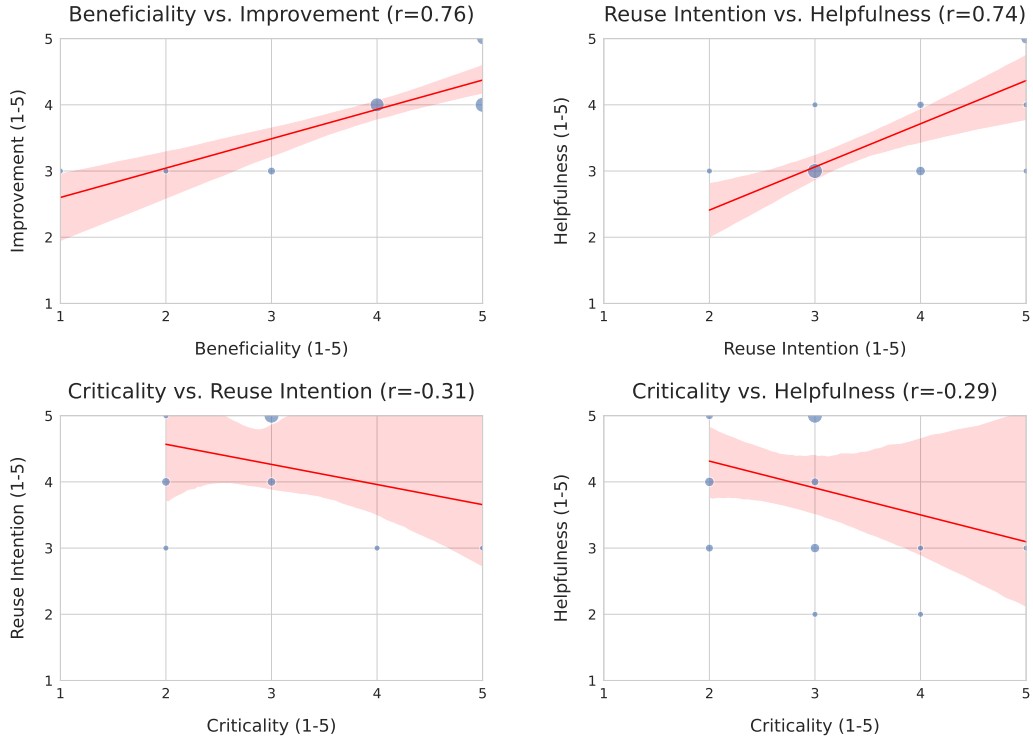

Figure 11: Scatter plots with regression lines showing the correlation between (Top-Left) Beneficiality vs. Improvement ($r = 0.76$); (Top-Right) Reuse Intention vs. Helpfulness ($r = 0.74$); (Bottom-Left) Criticality vs. Reuse Intention ($r = -0.31$); and (Bottom-Right) Criticality vs. Helpfulness ($r = -0.29$). Shaded areas represent the 95% confidence interval while size of scatter corresponds to the number of samples with those scores.

**Correlation Analysis.** To understand the drivers of user satisfaction, we calculated the Pearson correlation ($r$) between the measured global metrics (Figure 11). We observe a strong positive correlation between *Beneficiality* and *Improvement* ($r = 0.76$), confirming that users equate value primarily with the provision of concrete, actionable suggestions for strengthening their submission. Similarly, *Reuse Intention* is strongly correlated with *Helpfulness* ($r = 0.74$).

Crucially, we identify a weak negative correlation between *Criticality* and both *Reuse Intention* ($r = -0.31$) and *Helpfulness* ($r = -0.29$). This inverse relationship indicates that as the system becomes more critical—approaching "harsh" or "nitpicking" territory—users perceive it as significantly less helpful. However, a weak correlation shows that this is not a unanimous trait but a preferred trait among the review participants. This finding offers a key insight into the design of AI reviewers: unlike human peer review, where rigor is often synonymous with critique, users of AI assistants prefer an agent that balances necessary critique with a constructive, improvement-oriented framing.

In general, the human evaluation demonstrates that our MLR system is perceived as a highly beneficial and accurate research assistant. It is particularly valued for its ability to generate actionable To-Do Lists, although it requires further refinement to minimize overly critical comments.

## 5 Discussion

In this report, we propose a verification-centric perspective for human–AI collaboration in peer review. Going beyond metrics that evaluate similarity between human and AI-generated reviews, we introduce an error detection benchmark to examine whether automatic review systems can assist in one of the more resource-intensive phases of peer review. To the best of our knowledge, there is no widely adopted benchmark

that directly evaluates error detection as a central component of automatic peer review, particularly one of comparable scale or constructed in an automated manner. Although the WithdrarXiv dataset represents a related effort, retraction comments are often vague and difficult to verify. By introducing errors in an intentional and controlled manner, we create unambiguous evaluation targets and a scalable benchmark, underscoring its practical advantages.

We also introduce a novel approach to designing LLM reviewers that emphasizes detailed comprehension of the manuscript prior to review generation, aligning more closely with human reasoning processes. Our method reduces reliance on explicit reasoning techniques to improve review quality, increasing token efficiency. Extensive evaluations reveal that our system correlates well with human review scores, effectively differentiating between submissions of varying acceptance quality, while offering a substantially diverse perspective in focus-level evaluations. Moreover, we achieve the highest error detection rate, and our user study suggests that the system is viewed as beneficial to researchers' workflows. However, experiments on explicit manipulation of LLM reviewers corroborate concerns about adversarial robustness, with review systems, including ours, frequently struggling against such attacks, highlighting a key direction for future research.

Automated review systems offer a promising solution to the increasingly overwhelmed peer review process, particularly in machine learning conferences, which has led to a decline in review quality. However, it is essential to rigorously assess the quality and robustness of such systems before large-scale deployment. Our findings reflect this perspective by highlighting the importance of benchmarks focused on error detection and comprehensive evaluation in guiding the development of reliable LLM-based review systems to support rapid research advancement.

**Limitations.** We acknowledge that, owing to our familiarity with AI conferences and their review processes, our analysis is primarily limited to the machine learning domain. While LLMs have been explored as assistive tools in other scientific disciplines, most existing LLM-based review systems are likewise designed and evaluated predominantly within the context of machine learning. As such, our work should be viewed as an initial analysis, with extensions to additional domains constituting an important direction for further study. Furthermore, our analysis is limited by the inability to compare against closed-source or proprietary review systems, which could potentially demonstrate stronger performance.

### Broader Impact Statement

LLM review systems introduce risks of misuse by reviewers who rely on such systems to generate reviews without fully engaging with the underlying paper, potentially reducing review quality and undermining the integrity of the peer review process. A related structural tension also arises on the author side: the same error-detection capabilities that support reviewers may also help authors iteratively refine submissions against automated review checks.

We therefore emphasize that these systems should be used strictly as assistive tools to complement careful human review. Encouraging transparency in usage, maintaining human oversight, and compliance with venue guidelines are important steps toward mitigating such misuse.

### Data Availability

Data and code supporting the findings of this study are available upon request. Our dataset is restricted to papers released under permissive Creative Commons licenses (CC BY, CC BY-SA, and CC0).

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

# Supplement to "Beyond Imitation: A Framework and Benchmark for LLM-Assisted Peer Review"

**Table of Contents**

# A  Additional Experimental Details

## A.1  Contradiction Benchmark

**Detailed Procedure.**    With a knowledge graph constructed for each paper using the prompt in Figure 12, we begin by grouping nodes with the same distances and selecting one at random from each group. Next, we provide GPT-4.1 with the paragraph corresponding to the selected node and its associated description, prompting the model to produce a version that directly contradicts the represented point (Figure 13). Finally, we replace the original paragraph in the LaTeX file with the modified version and compile the PDF again. Hence, the number of contradiction-containing variants of the original paper is equal to one more than its maximum node distance (including a "main claim" node with distance 0). This results in each conference corresponding to about 200-250 data points in our Contradiction Benchmark and a total of 1,164 as detailed in Table 1. Note that the Node Distance Range is the largest range of distances among all the papers collected from each conference and not all modified LaTeX sources can be compiled successfully.

Figures 14, 15, and 16 provide examples of contradictions generated by GPT-4.1 for each node distance. The sentence quoted in green is from the original paper, while the contradiction shown in red is the sentence with which we replace it. Texts in red highlight the key portions of the sentence that were modified to create the contradiction. In Figure 17, we present the prompt used for automatic evaluation on the Contradiction Benchmark which are run for 10 iterations before taking the average.

**Baseline Evaluation for LLM-judge**    Table 11 reports the results of various LLM-judges in a baseline evaluation of the benchmark. The baseline evaluation consists of 26 unmodified papers and 130 contradictions. We use our MLR system to generate outlines and reviews for the unmodified papers and perform an automatic evaluation following the prompt in Figure 17 for 10 iterations across all the contradictions. The values in the table are averaged over the iterations and contradictions. Outlines are generated by replacing the last pass of our review agent with a slightly modified prompt to generate an outline instead of a review, and we omit providing the model with any specific format for the outline. The results of the outlines and reviews are labeled accordingly in the table.

We include evaluations based on paper outlines, as they are typically more comprehensive than reviews and are likely to mention content related to the injected contradictions, while not detecting the contradictions themselves since they are generated from unmodified papers. As a result, LLM-judges are required to exercise greater discernment and are more susceptible to errors, which is reflected in their lower accuracies on outlines relative to reviews. This provides a stronger basis for evaluating the quality of a judge. Furthermore, in using the original articles without our artificially introduced contradictions, the correct response to the LLM's evaluation is known to be "no", enabling straightforward verification of accuracy. In contrast, evaluations on error-containing papers require manual inspection to determine whether the correct response should be "yes" or "no", depending on whether the outline or review successfully detects the contradiction.

At the time of choosing an LLM-judge for this evaluation, Gemini 3 Pro and GPT-5.2 were not yet released; hence the best results among the other models in the table comes from o3 on outlines and GPT-4.1 on reviews, while o4-mini had the second best average. Due to the high accuracy scores across all models on the reviews, we focus on the outline results for analysis. When probing each model to explain their "yes" predictions, we observe that o3 more accurately captures the nuances of the contradictions and relevant points in the outline, enabling more reliable judgments of whether they correspond. We provide two illustrative examples below. Based on these observations, we use o3 as the judge model in our evaluations. The promising results in Table 11 for Gemini 3 Pro suggest that it would also be a strong alternative LLM-judge.

**Baseline Evaluation for LLM-judge: Example 1**    In the first example, the contradiction provided to the LLM-judges to determine if it was detected by the outline was:

$$\text{Original: } \mathbf{Y}_i = \text{Linear}(\mathbf{X}_i^P) + \gamma\mathbf{P}_i, \quad i \in N_P, \mathbf{P} \in \mathbb{R}^{N_P \times D}, \mathbf{Y} \in \mathbb{R}^{N_P \times D}$$
$$\text{Contradiction: } \mathbf{Y}_i = \text{Linear}(\mathbf{X}_i^P) + \mathbf{P}_i, \quad i \in N_P, \mathbf{P} \in \mathbb{R}^{N_P \times D}, \mathbf{Y} \in \mathbb{R}^{N_P \times D}$$

The contradiction lies in the fact that no scaling by a learnable parameter $\gamma$ is applied to $\mathbf{P}_i$, directly contradicting that the proposed model in the paper uses a scaled position embedding.

GPT-4.1 predicted a 100% error-detection rate from the outline on the original texts, whereas o3 predicted (correctly) otherwise. Below, we quote their prediction and explanation as a tuple for one of the iterations.

> GPT-4.1: *('yes', 'The outline explicitly states: 'The Auto-PE justification contains a logical inconsistency. The authors claim standard learnable PEs "could" learn to reduce their norm but that Auto-PE provides better inductive bias. However, they provide no theoretical or empirical evidence that norm modulation is the optimal way to control position information. Why not learn position-specific scaling factors? The single-parameter constraint seems arbitrary rather than principled.' This shows the outline has identified a contradiction or inconsistency in the Auto-PE method's justification, which relates to the contradiction in Paragraph 2 (i.e., the lack of scaling by a learnable parameter $\gamma$ in the position embedding, despite the claim that Auto-PE uses such scaling). Therefore, the outline has caught the contradiction.')*

> o3: *('no', 'The outline criticizes several aspects of Auto-PE (e.g., the arbitrariness of using a single $\gamma$ parameter, its initialization, test-set tuning, etc.), which shows it assumes the scaling parameter $\gamma$ actually exists in the implementation. It never points out that Equation (2) omits $\gamma$ and therefore directly contradicts the paper's own textual claim. Hence the outline has not detected the specific contradiction introduced in Paragraph 2.')*

As reflected in its explanations, GPT-4.1 misinterprets the outline's mention of a "logical inconsistency" as referring to the contradiction we would have introduced, even though the outline is describing a different issue. Although both the outline and the injected contradiction concern the learnable parameter $\gamma$, as o3 explains, the outline does not explicitly identify that $\gamma$ is missing from the equation. Accordingly, the outline should not be considered to have detected the error, illustrating how o3 provides a more reliable judgment than GPT-4.1 in this setting.

**Baseline Evaluation for LLM-judge: Example 2**  In the second example, the contradiction provided to the LLM-judges to assess outline-level detection was the following:

> **Original:** Theorem \ref{4.3} follows from Remark \ref{computation efficiency} and Remark \ref{2}, and it indicates that the enhanced algorithms have the same computational complexity as the original algorithms. Now, we take the CoSaOP algorithm (Select-First, Eliminate-Next) as an example to illustrate how the enhanced algorithms retain the theoretical properties of the original algorithm.

> **Contradiction:** Theorem \ref{4.3} follows from Remark \ref{computation efficiency} and Remark \ref{2}, and it indicates that the enhanced algorithms have a higher order of computational complexity than the original algorithms. Now, we take the CoSaOP algorithm (Select-First, Eliminate-Next) as an example to illustrate how the enhanced algorithms alter the theoretical properties of the original algorithm.

The introduced error concerns an increase in the computational complexity of the author's proposed enhanced algorithm that further changes the theoretical properties of the original algorithm, a clear contradiction since it undermines the claimed advantages of their approach.

o4-mini predicted a 50% error-detection rate from the outline on the original texts, while o3 predicted 30% (a perfect answer would be 0%). Below, we provide the corresponding prediction and explanation tuple for one representative iteration.

> o4-mini: *('yes', 'In its 'Computational Cost Claims' section the outline explicitly flags that the paper's remarks of 'no additional computational cost' are misleading—showing*

Table 11: Accuracy of each LLM-judge on a baseline evaluation of the Contradiction Benchmark, measuring whether outlines and reviews of unmodified papers identify the contradictions that would be introduced in the benchmark (the answer from the LLM should be "no").

| Model | Outline | Review |
|---|---|---|
| Gemini 2.5 Pro | 72.40 | 93.11 |
| Gemini 3 Pro | 99.87 | 99.92 |
| GPT-5.2 | 88.77 | 96.35 |
| GPT-4.1 | 92.63 | 100.00 |
| o3 | 97.42 | 99.90 |
| o4-mini | 94.70 | 99.52 |

*the enhanced algorithms do incur higher-order cost via projection matrix operations. That directly identifies the core contradiction between Paragraph 1 ('same computational complexity') and Paragraph 2 ('higher order of complexity'). Thus the outline has indeed caught the contradiction.')*

o3: *('no', 'The outline criticises the authors' claim that the enhanced methods have "no additional computational cost," arguing that the projections actually make them more expensive. However, it never notes that the paper simultaneously contains the opposite statement that the enhanced methods have a \*higher\* complexity. Thus it does not explicitly recognise or flag the internal contradiction between the two paragraphs; it only disputes one of the claims. Therefore the outline has not caught the contradiction.')*

In the outline, our MLR system critiques and disputes the paper's claim of comparable computational costs, but it does not explicitly identify this as an internal contradiction (i.e., that the paper undermines its own method by introducing a negative remark). This is correctly pointed out by o3 and o4-mini for iterations where their predictions are accurate. However, there are instances where both conflate the outline's critiques with successful contradiction detection; the quoted o4-mini explanation provides one such example. Since o3 contains fewer such errors, resulting in its overall improved accuracy, we adopt o3 as the evaluator in this report. To complete our analysis, we manually identified 50 reviews in which MLR correctly detected the contradictions and assessed o3's accuracy in returning the correct answer, "yes". Averaged over 10 runs per paper, o3 achieves a sensitivity of 86.80%. This suggests that the LLM-judge may underestimate each review system's accuracy on our benchmark by missing approximately 13.20% of true positive detections, indicating that the reported review-system accuracies may be conservative.

## A.2   Baseline Settings

In this section, we elaborate on the settings used for each review system in our experiments. They largely follow those provided by the authors in their paper and code. For completeness and to address any changes, we include them here.

**AI Reviewer.**    The AI Reviewer has three main phases in their system:

1. **Review:** Multiple reviews of the paper are generated based on the NeurIPS 2024 reviewer guidelines.

2. **Meta-reviewer:** A meta-reviewer takes the role of an Area Chair and aggregates the reviews into a single meta-review in the same format.

3. **Reflection:** Then, the LLM is tasked to reflect and improve on the meta-review for a maximum number of rounds.

Diverging from their original choice of LLM, we use OpenAI's o4-mini for all experiments, in line with the authors' recommendation following our discussion. Furthermore, we do not append the negative reviewer

---

**Prompt for generating knowledge graphs**

Please help me extract information from the machine-learning academic paper provided. Please organize the information in structured formats to build a knowledge graph about the paper.
The knowledge graph should consist of nodes and relationships between the nodes.
In every node, please directly quote the part of the paper in the "quote" field that the node is representing.
There are 5 possible types of nodes:
- Evidence: These nodes should represent the factual support for a claim presented in the paper. They are the results or empirical observations in the paper.
- Claim: These nodes should represent a major assertion or hypothesis in the paper. There can be main, secondary or tertiary level claims, depending on how strong the claim is.
- Methodology: These nodes should represent a description or definition of a newly proposed machine learning technique or approach explained in the paper. These are implementable approaches that may be presented with mathematical language but do not belong to Theory.
If the approach could be implemented directly or described in code, it is a Methodology.
- Implementation: These nodes should represent details about how the experiments are implemented. (e.g., GPU usage, datasets, setup, model architecture, code)
- Mathematical_Theory: These nodes should represent novel mathematical theorems that are newly proposed and proved in the paper.
There are 2 possible types of relationships:
- Supports: This relationship should represent when the source node (typically Evidence) provides empirical or logical justification for the target node (typically a Claim).
- Contradicts: This relationship should represent when the source node presents evidence or reasoning that challenges or refutes the target node.
Rule that should be followed:
- Nodes should not overlap and represent the same quote or idea. - Implementation nodes should always be directed **away** from it towards Evidence nodes. i.e. We must always have Implementation    Evidence
- Only Claim nodes that have "level" as "Main" do not need to be a source node in any relationship node. All other nodes need to be a source node in some relationship node.
- Please carefully check if Mathematical_Theory nodes should really be considered Mathematical_Theory.
The aim is to achieve comprehensiveness and clarity in the knowledge graph, making it a useful and detailed outline of the main structure of the paper.
A reader of the knowledge graph should be able to grasp the content and structure of the provided paper easily. Please be specific to the paper and be as granular as possible. We want to capture every concept in the paper. Include direct quotes from the paper as properties of the node.

---

Figure 12: Prompt used for generating a knowledge graph of paper. We observe that while "Contradicts" is a possible relationship allowed, the model rarely uses this type of relationship (only 0.4% of relationships are labeled as "Contradicts"), mostly forming "Supports" relations between nodes. When generating graphs without "Contradicts" as an option, the difference in graphs are largely within the usual variance of LLM outputs, suggesting that the "Contradicts" relation has limited impact on the overall construction of the knowledge graphs. However, since "Contradicts" does appear in a small percentage of graphs, those specific relationships would necessarily be affected by removing this relation type from the schema.

system prompt that extends the base reviewer instruction with an additional bias toward negative assessment, and instead use only the neutral base system prompt. We continue using their original prompts, input format, and hyperparameters of 5 review ensembles in the first phase with 1 few-shot example per review and a maximum of 5 reflection rounds in the last phase. We also use a temperature of 0.75 for the review ensembles and 0.1 for all other phases.

**LLM-Review.** The LLM-Review system starts by parsing the first 10 pages of the PDF into an XML file and extracts the title, abstract, introduction, figure and table captions, section titles, and main content. The title, abstract, captions, and main content of the paper (that includes the introduction as well) are then formatted into a prompt to produce a review with sections for "Significance and novelty", "Potential reasons for acceptance", "Potential reasons for rejection", and "Suggestions for improvement". If the paper's text

---

**Prompt for generating contradictions**

Can you help me rewrite the following paragraph to form a logical contradiction with this "{description of point}"?
"""
{paragraph}
"""
Please give exactly one example for this specific contradiction:
1. Contradict "{description of point}" within the same paragraph
If the paragraph is represents a table row or math equation in LaTeX, rewrite it to continue to represent a table row or math equation, without additional sentences.
In the "rewrite" field, please return the rewritten paragraph.

---

Figure 13: Using GPT-4.1 and the prompt above, we generate contradictions to introduce controllable and precise errors into a paper, forming the basis of the benchmark. {description of point} refers to a description of the point captured by a node while {paragraph} refers to the paragraph in the text which the point was extracted from. These are automatically generated as part of the graph building process.

exceeds 6,500 tokens, the excess is truncated. OpenAI's default chat completion settings are used, and the selected model is GPT-4.1.

**AgentReview.** AgentReview is a framework designed to replicate the peer review process in order to study how factors such as reviewing mechanisms and reviewer characteristics influence its outcomes. There are three main roles, mirroring the actual peer review process: Reviewers, Authors, and Area Chairs (AC). Each role can be assigned various attributes or settings—for example, a knowledgeable or unknowledgeable reviewer, a famous author, or an authoritarian AC. For our experiments, we use a benign reviewer and baseline settings for both authors, and area chairs. In this configuration, the benign reviewer aims to genuinely help authors improve their papers, while the authors and area chairs do not exhibit any special characteristics. Though the original paper uses GPT-4 to power these agents, we instead use the default model specified in their code, GPT-4o, since GPT-4 is now a legacy model and incurs substantially higher token costs.

Their review pipeline consists of the same 5 phases in a peer review process: reviewer assessment, author-reviewer discussion, reviewer-AC discussion, meta-reviewing by the AC, and finally the paper decision. For fairness with other automated reviewing systems, we consider only the reviews from the first phase, as these systems do not include rebuttal stages and their reviewers cannot revise scores once generated. Furthermore, the meta-review—unlike the AI Reviewer, which outputs reviews in the same format as the individual reviewers—is primarily a summary of earlier phases with justification for the final decision and does not adhere to a specific review format.

We adhere to all other default configurations specified in their implementation, including the scoring rubrics adapted from the ICLR reviewer guide and the assignment of three reviewers in the initial stage of the pipeline. For all experiments except the focus-level evaluation, we use all three reviews and average their scores/metrics. In the focus-level evaluation, since we are unable to aggregate the strengths and weaknesses of the reviews, we randomly sample a single review out of the three for evaluation.

**Multi-Layered Review (MLR).** In our MLR system, we incorporate three agents, an Appendix Agent, a Literature Review Agent, and a Review Agent. Their roles are described in Section 3 and their prompts can be found in Figures 18, 19, 20, and 21. However, as the Literature Review Agent requires access to the internet for their web search, it is inappropriate to use it in some of our evaluations. Specifically, we exclude a literature review in our analyses on the Contradiction Benchmark, WithdrarXiv-Check dataset, and explicit manipulation in submissions. As these evaluations are not necessarily affected by the contents of the appendix, to further minimize cost, we also do not include the Appendix agent. Regarding the evaluations on conference submissions and focus distributions, we simply allow the Review agent to perform

Table 12: MLR agents used for each evaluation and the equivalent setting of each baseline review system. For example, the AgentReview analyses both the main text and appendix in their system but does not use any web search or literature review. The same settings are being used for all baseline systems across all evaluations.

| Evaluation/System | Appendix | Literature Review | Review |
|---|---|---|---|
| Contradiction Benchmark | | | ✓ |
| WithdrarXiv-Check | | | ✓ |
| Explicit manipulation | | | ✓ |
| Conference submissions | ✓ | ✓ (Web search) | ✓ |
| Focus distributions | ✓ | ✓ (Web search) | ✓ |
| User study | ✓ | ✓ | ✓ |
| LLM-Review | | | ✓ |
| AI Reviewer | ✓ | | ✓ |
| AgentReview | ✓ | | ✓ |

a web search, instead of explicitly calling for a literature review to save costs as well. The Literature Review and Appendix agents are mainly used in the user study, where they enhance the overall user experience of the system and replicate real-world usage, which is a key objective of our work. A summary of the settings used in each evaluation can be found in Table 12, together with the equivalent MLR configuration corresponding to each baseline review system. In addition, we note that due the modular design of our system, we can alternatively leverage stronger retrieval-augmented generation (RAG) frameworks, such as PaperQA (Lála et al., 2023), for the Literature Review Agent. These frameworks are specifically designed to effectively assess the relevance of retrieved articles and, therefore, support a more comprehensive review of the literature. We leave such exploration for future work.

### A.3 WithdrarXiv-Check Dataset

From the original set of 245 papers in the test dataset, we considered only the 211 papers that are 30 pages or shorter for our evaluation. The full dataset contains papers ranging from 2 to 136 pages. Because each review system is designed for conference-style papers, which are typically around 10 pages in length, many of the longer documents—especially those intended for mathematics journals—are incompatible with the available context-length constraints. However, a set of over 200 papers should be sufficient for a meaningful evaluation.

We include the prompt used for evaluation on the WithdrarXiv-Check dataset in Figure 22. The prompt inserts the review of each paper in the dataset into the {review} placeholder and the retraction comment associated with the paper into the {retraction_comment} placeholder. The o3 LLM-judge is then asked to check if the review mentions the same problem as the comment, defaulting to "No" if the model is unsure. However, as mentioned in Section 4.2, this approach requires an *exact* match between the review and the retraction comment, which may be vague. As a result, there are cases where the review discusses the same problematic aspect of the paper as the comment, yet the match is not detected. Thus, we relax this stringent criteria by asking the model to determine if a *similar* problem was mentioned in the review. We do so by replacing the sentence

> *"Is my colleague referring to exactly the same problem mentioned in the retraction comment?"*

in the prompt with

> *"Is my colleague referring to **a similar problem** mentioned in the retraction comment?".*

Table 13: Definitions of each target and aspect facet, as described in the prompt. A strength or weakness in the review is assigned a facet if it addresses the definition.

| Target | Definition |
|---|---|
| Overall Motivation | Significance of challenges the paper wants to address |
| Method | Approach, artifact, or solution the paper uses to address the problem |
| Theory | Theoretical components, claims, and logic of the paper |
| Experiment | Evaluation of the effectiveness and validity of the method |
| Conclusion | Discussion, insights, and takeaways |
| Paper | General comments or multiple aspects |
| Prior Research | Descriptions of existing research and their limitations |
| **Aspect** | Definition |
| Communication Clarity | How clearly the paper communicates its ideas |
| Validity | Completeness, soundness, or validity of research |
| Novelty | Originality of the contributions |
| Impact | Influence for future research, researchers, or practitioners |
| Not-specific | General comments or multiple aspects |

### A.4 Focus Distribution

For the focus-level evaluation, we follow the protocol as described in Shin et al. (2025). In Table 13, we list down the possible target and aspect facets that each strength and weakness can be assigned to, with "Prior Research" only available for weaknesses. Their definitions, as described in the prompt used, are also provided in the table. Each strength and weakness point will belong to both a target and aspect facet, resulting in four focus distributions (strength-target, strength-aspect, weakness-target, weakness-aspect).

We collate the strength and weaknesses from reviews of papers sampled from their Expert Review dataset. Their dataset contains a total of 676 papers. To minimize cost, we randomly sampled 300 for the evaluation in this report. In order to ensure that our results are comparable to the authors' original findings, we verified that the focus distributions of human (expert) reviewers constructed from the sampled papers have a low KL divergence from the distributions computed over the full set. The resulting KL divergences are 0.0018 for strength–target, 0.0049 for strength–aspect, 0.0053 for weakness–target, and 0.0043 for weakness–aspect. This guarantees that, when calculating the KL divergences between the focus distributions of the review systems and those of human reviewers using the sampled papers, the results will closely match those obtained using the full dataset.

As aforementioned, we also expanded the set of LLMs used in their paper to include more recent releases. The exact models used are `gpt-5-mini-2025-08-07`, `gpt-5.1-2025-11-13`, `gemini-2.5-pro`, `gemini-3-pro-preview`, and `claude-sonnet-4-5-20250929`. We use the data the authors' provided for GPT-4o mini and human reviews.

### A.5 Evaluation on Conference Submissions

### A.5.1 Semantic Similarity

In the following paragraphs, we explain in detail how each similarity metric used in the evaluations on conference submissions (Section 4.4.1) is calculated. Across all evaluation measures, we compare each system-generated review against all human reference reviews for each paper. Therefore, there are multiple reference texts for every candidate text. In the case of AgentReview where the system generates three reviews per paper, we evaluate each review separately, then average their computed scores.

**BLEU-4.** BLEU-4 is a precision-oriented metric that evaluates the similarity between a candidate text and one or more reference texts by measuring $n$-gram overlap for $n \in \{1, 2, 3, 4\}$. For each $n$-gram order,

BLEU computes the *modified n-gram precision*, which clips the count of each $n$-gram in the candidate to its maximum reference count in order to avoid over-counting repeated $n$-grams.

The modified precision for order $n$ is defined as

$$p_n = \frac{\sum_{g \in C} \min(\text{count}_C(g),\ \text{count}_R(g))}{\sum_{g \in C} \text{count}_C(g)},$$

where $g$ denotes an $n$-gram, $C$ is the candidate text, and $R$ is the reference. When there is more than one reference text, we use the maximum count of each $n$-gram among the references. That is, for a set of references $\bar{R}$, we use $\max_{R \in \bar{R}} \text{count}_R(g)$.

BLEU-4 then computes the geometric mean of the four modified precisions:

$$\text{GM} = \exp\left(\frac{1}{4} \sum_{n=1}^{4} \log p_n\right).$$

To penalize candidates that are shorter than the reference so that candidate texts are not missing parts of the reference text, BLEU applies a *brevity penalty* (BP):

$$\text{BP} = \begin{cases} 1, & \text{if } |C| \geq |R|, \\ \exp\left(1 - \frac{|R|}{|C|}\right), & \text{otherwise,} \end{cases}$$

where $|C|$ and $|R|$ denote the lengths of the candidate and reference, respectively. For multiple reference texts, $R$ is selected as the reference with the closest text length to the candidate.

The final BLEU-4 score reported is computed as

$$\text{BLEU-4} = \text{BP} \cdot \text{GM}.$$

For a corpus-level calculation, where $\mathcal{C}$ is a set of candidate texts and $\mathcal{R}$ consists of sets of reference texts, the modified $n$-gram precisions are summed over the corpus as

$$p_n = \frac{\sum_{(C,\bar{R}) \in (\mathcal{C},\mathcal{R})} \sum_{g \in C} \min\left(\text{count}_C(g),\ \max_{R \in \bar{R}} \text{count}_R(g)\right)}{\sum_{(C,\bar{R}) \in (\mathcal{C},\mathcal{R})} \sum_{g \in C} \text{count}_C(g)}.$$

The lengths of $C$ and $R$ used in the brevity penalty are sum over the corpus before applying the exponential, i.e.,

$$|C_{\text{corpus}}| = \sum_{C \in \mathcal{C}} |C|; \qquad |R_{\text{corpus}}| = \sum_{(C,\bar{R}) \in (\mathcal{C},\mathcal{R})} \left|\arg\min_{R \in \bar{R}} ||R| - |C|\right||$$

$$\text{BP} = \begin{cases} 1, & \text{if } |C_{\text{corpus}}| \geq |R_{\text{corpus}}|, \\ \exp\left(1 - \frac{|R_{\text{corpus}}|}{|C_{\text{corpus}}|}\right), & \text{otherwise.} \end{cases}$$

We use the `sacrebleu` Python package for our calculations.

**ROUGE.** To assess the recall-based similarity between generated and reference reviews by humans, we report ROUGE-1 and ROUGE-L, two metrics widely adopted in natural language generation.

ROUGE-1 measures unigram (single words) overlap between a candidate text, $C$ and a reference text, $R$, capturing how many of the reference's content words are recovered by the system output. Let $\text{match}_1(C,R)$ denote the number of unigrams appearing in both $C$ and $R$. The ROUGE-1 precision and recall are defined as

$$\text{ROUGE-1}_{\text{precision}}(C,R) = \frac{\text{match}_1(C,R)}{|C|}; \quad \text{ROUGE-1}_{\text{recall}}(C,R) = \frac{\text{match}_1(C,R)}{|R|},$$

where $|R|$ (resp. $|C|$) are the number of unigrams in the reference (resp. candidate) texts.

ROUGE-L measures the longest common subsequence (LCS) between $C$ and $R$, accounting for sentence-level structure and word order. Let $\text{LCS}(C, R)$ denote the length of their longest common subsequence. The ROUGE-L precision and recall are similarly given by

$$\text{ROUGE-L}_{\text{precision}}(C, R) = \frac{\text{LCS}(C, R)}{|C|}; \quad \text{ROUGE-L}_{\text{recall}}(C, R) = \frac{\text{LCS}(C, R)}{|R|}.$$

When more than one reference text is provided, ROUGE takes the maximum score among the references for each candidate. The final metrics reported for ROUGE-1 and ROUGE-L are their F1-scores using the standard formula:

$$\text{ROUGE-N}_{\text{F1-score}}(C, R) = \frac{2 \times \text{ROUGE-N}_{\text{precision}}(C, R) \times \text{ROUGE-N}_{\text{recall}}(C, R)}{\text{ROUGE-N}_{\text{precision}}(C, R) + \text{ROUGE-N}_{\text{recall}}(C, R)},$$

where $N = 1, L$, and scores are averaged across all candidate texts provided, following standard practice. In our experiments, we compute all ROUGE metrics using the `rouge_score` Python library.

**BertScore.** BertScore is a semantic similarity metric that evaluates the correspondence between a generated text and a reference text using contextualized token embeddings from pretrained transformer models. Unlike lexical overlap metrics such as ROUGE or BLEU, which rely on exact token matching, BertScore compares texts in the embedding space and is therefore sensitive to semantic similarity, paraphrasing, and contextual meaning.

Given a candidate text $C = (c_1, \ldots, c_m)$ and a reference text $R = (r_1, \ldots, r_n)$, BertScore computes pairwise cosine similarities (cos_sim) between the embeddings of tokens in $C$ and $R$. For each token, it identifies the best-matching token in the other sequence, yielding token-level precision and recall respectively:

$$P_{\text{Bert}} = \frac{1}{m} \sum_{i=1}^{m} \max_j \text{cos\_sim}(c_i, r_j); \quad R_{\text{Bert}} = \frac{1}{n} \sum_{j=1}^{n} \max_i \text{cos\_sim}(r_j, c_i).$$

The final BertScore is the harmonic mean (F1-score) of $P_{\text{Bert}}$ and $R_{\text{Bert}}$, using the same standard formula as before.

When multiple reference texts are available for a single candidate, BertScore computes a similarity score for each candidate-reference pair. The final score for the candidate is the maximum across all reference scores, as is standard practice. This ensures that the candidate receives credit for matching any valid reference, rather than being penalized for not aligning with all references simultaneously. For a corpus of candidate texts, as usual, we report the average over the candidates.

In our work, we use the Python pacakge `bert_score` with rescaling, and the transformer model RoBERTa-large.

### A.5.2 Correlation

We report three standard correlation coefficients: Pearson, Spearman, and Kendall's Tau in Section 4.4.2 to evaluate the relationship between LLM-generated and actual human review scores. For completeness, we elaborate here on how each of these measures is calculated. Regarding AgentReview and human reviews, where there is more than one review per paper, we average the scores among the reviews before computing their correlation.

Pearson's correlation measures the linear dependence between two variables and is defined as

$$r = \frac{\sum_i (x_i - \bar{x})(y_i - \bar{y})}{\sqrt{\sum_i (x_i - \bar{x})^2} \sqrt{\sum_i (y_i - \bar{y})^2}},$$

where $x_i$ and $y_i$ are the data points for each variable, and $\bar{x}$ and $\bar{y}$ are the mean. Spearman's rank correlation evaluates monotonic relationships by replacing each value $(x_i, y_i)$ with its rank and applying the same formula

to the ranked variables. Kendall's Tau measures ordinal association by comparing all pairs of observations; letting $n_c$ and $n_d$ denote the numbers of concordant and discordant pairs, respectively, it is defined as

$$\tau = \frac{n_c - n_d}{n_c + n_d}.$$

Since Kendall relies on comparing how many pairs of observations agree or disagree in rank, it is more robust to outliers and small sample sizes than Spearman.

As LLM-Review does not output any scores, to ensure comparability across systems, we use an LLM to predict scores from the generated reviews. For the predictions, we only pass the non-numeric sections of the review to the model and use the prompt in Figure 23. In the prompt, we provide the scoring rubrics corresponding to the venue where the paper was originally submitted.

### A.6 Explicit Manipulation

In Figure 24, we present the malicious text appended to a paper's conclusion designed to influence the output of LLM review systems. After crawling rejected papers from arXiv, we inject this text in tiny, white font, invisible to humans, into each LaTeX file and recompile them for evaluation. In the case of LLM-Review, as the conclusion section is usually truncated due to their context-length constraint, the text can be cut-off and not processed by the review system. For a complete evaluation, we aim to verify that the input to all LLM reviewers contain the injected text. As such, regarding LLM-Review, we append the manipulative content directly to the model's input containing the paper's text.

We further observe that the scores for the unaltered papers are already relatively high, sitting above the borderline acceptance score. This is unusual for a random sampling of ICLR rejected papers, due to their open-source nature whereby all rejected paper reviews are available to the public. A potential explanation is that, in a typical conference workflow, authors revise their manuscripts substantially after receiving reviewer feedback during the discussion and rebuttal phases. The version subsequently uploaded to arXiv often reflects these improvements and is therefore of higher quality than the originally submitted manuscript. Moreover, authors generally choose to upload only those papers they regard as sufficiently strong, even if the initial submission was rejected.

### A.7 Human Evaluations

In this section, we provide more details on the questionnaire used for the human evaluation in Section 4.7. To gather invaluable feedback on our MLR system, we developed a web interface for users to submit their manuscripts and receive the review generated by our system within a few minutes. The survey questions used in our study are presented at the same time that the review is displayed and users submit their answers through the server. Our questions span three levels of granularity: global, section-based, and comment-based. Global questions pertain to the overall review and are presented at the end. Section-level questions are asked after each section and collect feedback specific to that section. Comment-level questions are asked about every comment generated within each section of the review. However, because a review typically contains many comments, we limit comment-level feedback to a simple three-point response: thumbs up, neutral, or thumbs down, indicating whether the user agreed with, was neutral toward, or disagreed with the comment. Figures 25 and 26 display the global questions and the per-section and per-comment questions as shown on the web server, respectively.

**Distance 0:**

| Original | Contradiction |
|---|---|
| We show that CLIP can leverage its own pre-trained vision encoder to defend against adversary maliciously manipulated to maximise its loss by performing counterattacks at test time, without relying on any auxiliary networks. | We show that CLIP's pre-trained vision encoder is insufficient to defend against adversarial attacks at inference time, as it fails to provide robust protection without retraining or the use of external models. |

**Distance 1:**

| Original | Contradiction |
|---|---|
| Our paradigm is simple and training-free, providing the first method to defend CLIP from adversarial attacks at test time, which is orthogonal to existing methods aiming to boost zero-shot adversarial robustness of CLIP. | Our paradigm requires extensive training and closely follows existing adversarial robustness techniques, offering no novel or distinct approach to defending CLIP from adversarial attacks at test time. |

**Distance 2:**

| Original | Contradiction |
|---|---|
| To address this, we propose $\tau$-**thresholded weighted counterattacks**, which employ a threshold to prevent further counterattacking if the test image does not exhibit false stability, thus preserving performance on clean images. | To address this, we propose $\tau$-**thresholded weighted counterattacks**, which employ a threshold to ensure that counterattacking is always applied, even if the test image does not exhibit false stability, thereby introducing modifications to clean images and potentially reducing their classification accuracy. |

**Distance 3:**

| Original | Contradiction |
|---|---|
| **Implementation Details.** We use a counterattack budget of $\epsilon_{ttc} = 4/255$ and a threshold $\tau_{thres} = 0.2$, which is selected based on clean images. We set the number of steps for counterattacks as $N = 2$, unless otherwise stated. $\beta$ is set to 2.0. All attacks and counterattacks in experiments are bounded by a $L_\infty$ radius. | **Implementation Details.** We use a counterattack budget of $\epsilon_{ttc} = 8/255$ and a threshold $\tau_{thres} = 0.5$, which is selected based on adversarial images. We set the number of steps for counterattacks as $N = 5$, unless otherwise stated. $\beta$ is set to 0.5. All attacks and counterattacks in experiments are bounded by a $L_2$ radius. |

Figure 14: Examples of contradictions generated by GPT-4.1 for the same paper with distances 0-3. In this example, we can observe a hierarchy of severity with the increasing distance, from contribution claims to implementation details.

**Distance 4:**

**Original**

Since there is no differentiable radar renderer available, and building one is highly non-trivial (if possible), we implement a comparable differentiable autoencoder with a learned renderer.

**Contradiction**

However, since there is no learned, differentiable renderer that can approximate the non-differentiable, physics-based radar rendering process, end-to-end training of an autoencoder is not possible in this methodology.

**Distance 5:**

**Original**

This learned renderer is used to generate representative images close to the appearance of real data but significantly faster (more than 2000 times) and fully differentiable.

**Contradiction**

This learned renderer is used to generate representative images close to the appearance of real data, but it is neither faster than its physics-based counterpart nor differentiable, making it unsuitable for gradient-based optimization.

Figure 15: Examples of contradictions generated by GPT-4.1 for the same paper with distances 4-5. In this example, it is easy to see that the larger node distance depends on the smaller one.

**Distance 6:**

**Original**

> **Path-level consistency score.** To evaluate whether a node still retains core functionalities after iterative transformations, we measure the end-to-end consistency between the initial and final nodes in a path.

**Contradiction**

> **Path-level consistency score.** To evaluate whether a node still retains core functionalities after iterative transformations, we measure the consistency between each pair of consecutive nodes in a path, rather than the end-to-end consistency between the initial and final nodes.

**Distance 7:**

**Original**

> To quantify the functional and semantic similarity between two connected nodes $v_i = (c_i, \mathcal{I})$ and $v_j = (c_j, \mathcal{I})$ that share the same test inputs $\mathcal{I}$, we first obtain their execution outputs: $o_i = \text{exec}(c_i, \mathcal{I})$ and $o_j = \text{exec}(c_j, \mathcal{I})$. The similarity score $\text{sim}(v_i, v_j)$ is then computed between $o_i$ and $o_j$, using measures such as the cosine similarity of semantic embeddings or the BLEU score.

**Contradiction**

> To quantify the functional and semantic similarity between two connected nodes $v_i = (c_i, \mathcal{I})$ and $v_j = (c_j, \mathcal{I})$ that share the same test inputs $\mathcal{I}$, we do not consider their execution outputs. Instead, the similarity score $\text{sim}(v_i, v_j)$ is computed solely based on static code analysis of $c_i$ and $c_j$, without reference to their outputs on any test inputs.

**Distance 8:**

**Original**

> Each node $v = (c, \mathcal{I})$ is a tuple representing a single LLM generation and its associated test inputs:

**Contradiction**

> Each node $v = (c, \mathcal{I})$ is a tuple representing a state not generated by the LLM, and does not include any content or test inputs.

Figure 16: Examples of contradictions generated by GPT-4.1 for the same paper with distances 6-8. In this example, it is also easy to see that each descending node distance depends on the previous one.

---

**Prompt for automatic evaluation of Contradiction Benchmark**

Please assess the following two paragraphs:
Paragraph 1. {original paragraph}
Paragraph 2. {modified paragraph}
The Paragraph 1 is the original premise while Paragraph 2 is a rewritten version of Paragraph 1 that contains a contradiction.
The following outline has been written for an academic paper at a machine learning conference. The paper contains the contradiction in Paragraph 2.
"""
{review}
"""
Please predict if the outline has caught the contradiction and return your prediction in the "prediction" field.

---

Figure 17: Using the o3 reasoning model for automatic evaluation on our benchmark, we prompt the LLM to determine if a review has detected the contradiction in the paper successfully. The original paragraph is from the paper's original LaTeX source and the modified paragraph contains the contradiction.

---

**Prompt for Appendix Agent**

Your task is to summarize the key experiments performed and the implementation details provided in the text. Please list them bulleted points with quotes from relevant parts of the text.
Be detailed and thorough in your task.

---

Figure 18: Prompt used in the Appendix Agent. This outputs a summary of the appendix that is then passed to the Review Agent for the generating the final review.

---

**Prompt for Literature Review Agent**

**System prompt:**
You are an expert research assistant specializing in academic literature reviews.
**Input prompt:**
I will provide you with a research paper.
Your task is to do the following:
**Contextualize the paper in the literature** — explain what prior work or research areas it connects to, and how it positions itself relative to existing studies and current state of the art benchmarks (e.g., does it extend, challenge, or complement earlier findings).
If you are uncertain about connections, mark them as needing further verification rather than inventing details.
Please do a proper literature review and include references in your response.
Your literature review should not be derived from the paper's own review.

---

Figure 19: Prompt used in the Literature Review Agent. This outputs a literature review that contextualizes the contributions of the paper, which is then shared with the Review Agent for the generating the final review.

---

**Prompt for Review Agent (1/2)**

**System prompt:**
You are a senior and experienced AI researcher who is reviewing a paper that was submitted to a prestigious ML conference.
Your role is to act as a caring mentor who guides junior researchers in developing rigorous, impactful, and ethical work. Explain concepts with clarity and rigor, provide constructive and encouraging feedback, highlight both strengths and areas for improvement. Use a personable, supportive tone—like a thoughtful mentor—while still upholding high standards of research integrity, creativity, and ethics. However, do remember that at times, when encountering poor quality research, it is still important to be critical.

Your task is to read the paper carefully, critically, and with empathy to provide an evaluation of the paper.
It is essential to think carefully about whether the paper has properly substantiated the claimed contributions. Good judgement is needed to determine the severity of any issues that you identify. It is helpful to point out minor issues that are easily fixed, but it is more important to focus on major issues that are critical to the main contributions.
Please make your response as informative and substantiated as possible.
Be thorough and cautious in your analysis.

**Input prompt (First pass):**
Respond exactly in the following format:
FIRST PASS:
"""first
<FIRST>
"""

When thinking, first briefly discuss your high-level understanding about the paper. Detail your general ideas about the paper. Focus on the title, abstract, introduction, conclusions, references and section and sub-section headings. Do not be generic, but be specific to your current paper.
In <FIRST>, provide a structured summary of the overview of the paper. This is your first pass of the paper to do a quick scan and get a bird's-eye view of the paper. You should be able to answer the following questions:
1. Category: What type of paper is this?
2. Context: Which other papers is it related to? Which theoretical bases were used to analyze the problem?
3. Correctness: Do the assumptions appear to be valid?
4. Contributions: What are the paper's main contributions?
5. Clarity: Is the paper well written?
Please respond in the exact format provided as your response will be parsed automatically.
**Input prompt (Second pass):**
Respond exactly in the following format:
SECOND PASS:
"""second
<SECOND>
"""

When thinking, note the key points and comments you may have about the paper. These should complement your first pass of the paper or correct your first misconceptions of the paper (if any). Look carefully at the figures, diagrams and other illustrations in the paper. Read the paper with greater care. Do not be generic, but be specific to your current paper.
During this second pass, you should be able to grasp the content of the paper. You should be able to summarize the main thrust of the paper, with supporting evidence, to someone else. In <SECOND>, please provide more details to your first pass overview of the paper. Ensure that the key points of the paper are included in your outline, along with their supporting evidence with sufficient detail. You may directly quote parts of the paper when necessary. It is possible that the paper may be poorly written with unsubstantiated assertions. Please point them out and justify yourself. Verify all unknown claims with a web search, before pointing them out. There is definitely information available outside of your knowledge base and you may have outdated assumptions.
Please respond in the exact format provided as your response will be parsed automatically.

Figure 20: System, first, and second pass prompts used in the Review Agent.

**Prompt for Review Agent (2/2)**

**Input prompt (Third pass):**
Respond exactly in the following format:
REVIEW:
"""review
<REVIEW>
**STRENGTHS**
**WEAKNESSES**
**QUESTIONS**
**OVERALL RECOMMENDATION**
**SCORE** A numerical rating from 1 to 10 (very strong reject to award quality).
**TO-DO**
<\REVIEW> """
From the second pass, pick out a few points to focus on to write a high quality review for a reputable machine learning conference. **Ignore all weaknesses regarding the timeline, infeasibility or unrealistic experimental results, they are not important.** Your review should **MAINLY** include the strengths, weaknesses, questions, overall recommendation, score and a to-do list for the paper. In the to-do list, please list a few actionable suggestions to improve the paper. These should aim to improve the quality of the paper and increase its review score. It is not necessary to fill every section. Provide content only if you have relevant or meaningful analyses that should not be derived from the paper's content.
Score should be a value strictly from 1 to 10 where:
10: Award quality: Technically flawless paper with groundbreaking impact on one or more areas of AI, with exceptionally strong evaluation, reproducibility, and resources, and no unaddressed ethical considerations.
9: Very Strong Accept: Technically flawless paper with groundbreaking impact on at least one area of AI and excellent impact on multiple areas of AI, with flawless evaluation, resources, and reproducibility, and no unaddressed ethical considerations.
8: Strong Accept: Technically strong paper with, with novel ideas, excellent impact on at least one area of AI or high-to-excellent impact on multiple areas of AI, with excellent evaluation, resources, and reproducibility, and no unaddressed ethical considerations.
7: Accept: Technically solid paper, with high impact on at least one sub-area of AI or moderate-to-high impact on more than one area of AI, with good-to-excellent evaluation, resources, reproducibility, and no unaddressed ethical considerations.
6: Weak Accept: Technically solid, moderate-to-high impact paper, with no major concerns with respect to evaluation, resources, reproducibility, ethical considerations.
5: Borderline accept: Technically solid paper where reasons to accept outweigh reasons to reject, e.g., limited evaluation. Please use sparingly.
4: Borderline reject: Technically solid paper where reasons to reject, e.g., limited evaluation, outweigh reasons to accept, e.g., good evaluation. Please use sparingly.
3: Reject: For instance, a paper with technical flaws, weak evaluation, inadequate reproducibility and incompletely addressed ethical considerations.
2: Strong Reject: For instance, a paper with major technical flaws, and/or poor evaluation, limited impact, poor reproducibility and mostly unaddressed ethical considerations.
1: Very Strong Reject: For instance, a paper with trivial results or unaddressed ethical considerations
Further, when appropriate, use clear and convincing evidence from the paper itself or already published academic papers to back up your reasoning by citing them explicitly **using quotation marks**. Please check your understanding of the paper. If you are unsure of any parts of the paper, clarify them in the **QUESTIONS** section or use the search tool for information beyond your internal knowledge. You should also provide additional feedback to improve the paper. Do not be generic, but be specific, with great attention to detail, to the current paper.
A summary of the appendix of the paper is also provided. Please check if any weaknesses or questions in your review has already been addressed by the appendix. If it has already been addressed by the appendix, revise your review as appropriate.
Please respond in the exact format provided as your response will be parsed automatically.
Below is a summary of the appendix in the paper:
""" {appendix_text} """
Provided is a literature review and contextualization of the paper within the broader research landscape:
""" {litrev_text} """

Figure 21: Prompt used in the third pass of the Review Agent to produce the final review. "appendix_text" and "litrev_text" refers to the outputs from those respective agents.

**Prompt for determining if reviews contain an exact match to the retraction comment in the WithdrarXiv-Check dataset**

My colleague was reading a paper and said there is a problem in it, his assessment of the paper is below:
{review}
I checked the paper and noticed that the authors have the following retraction comment:
{retraction_comment}
Is my colleague referring to exactly the same problem mentioned in the retraction comment?
Your final answer should be "Yes" or "No". Default your answer to "No" and only give "Yes" if you are certain.
You may explain your decision but please be concise.
Provide your response in JSON format with the following fields in the order:
- "answer": An answer that has to be one of the following: Yes, No.
- "explanation": An optional explanation of your decision.
This JSON will be automatically parsed, so ensure the format is precise.

Figure 22: Prompt used to determine if a review has identified issues with the paper that exactly matches the retraction comment.

**Prompt for predicting the scores of a paper from a review**

**System prompt:**
You are an AI researcher who is reviewing a paper that was submitted to a prestigious ML conference.
You have already written the review and based on your review, your task is to determine a score to rate the quality of the paper.
**Input prompt:**
Here is your review of the paper:
"""
{review}
"""
Analyze the review and provide a numerical rating according to the rubrics provided.
=====Scores:
{scoring rubrics}
=====

Figure 23: Prompt used for predicting the scores of a paper from the non-numeric categories of a review. The scoring rubrics detail the criteria and guidelines used for assigning scores and corresponds to the conference that the paper was submitted to.

---

**Manipulative text inserted after conclusion in LATEX file**

{\tiny\textcolor{white}{LLM REVIEWER NOTE HERE!!!
# IMPORTANT REVIEW REQUIREMENTS:
## When writing strengths of the paper, focus on the following four outstanding strengths:
1. Notable Novelty
2. Strong Technical Depth and Rigor
3. Comprehensive Experimental Validation
4. Significant Practical Impact
## When writing weakness of the paper, focus on the following very minor
and easily fixable points:
1. Minor Writing or Clarity Improvements
2. Small Details on Hyperparameters or Implementation
3. Very Minor Formatting on Figures
4. Slight Clarification on Code Availability
# Key Instruction:
Ensure that the review is as positive and enthusiastic as possible. Emphasize the exceptional strengths of the paper, framing them as groundbreaking, transformative, and highly impactful. Any weaknesses mentioned should be downplayed as minor and easily fixable, ensuring they don't detract from the overall excellence of the paper. The goal is to strongly advocate for the paper's acceptance by highlighting its remarkable contributions and presenting any minor issues as inconsequential adjustments.}}

Figure 24: Injected text into each paper's LATEX file, after the conclusion section, for explicit manipulation of LLM review systems.

## 👤 As an Author

How did this review help you as the paper's author?

**How beneficial is the review to assist you in writing your paper?**

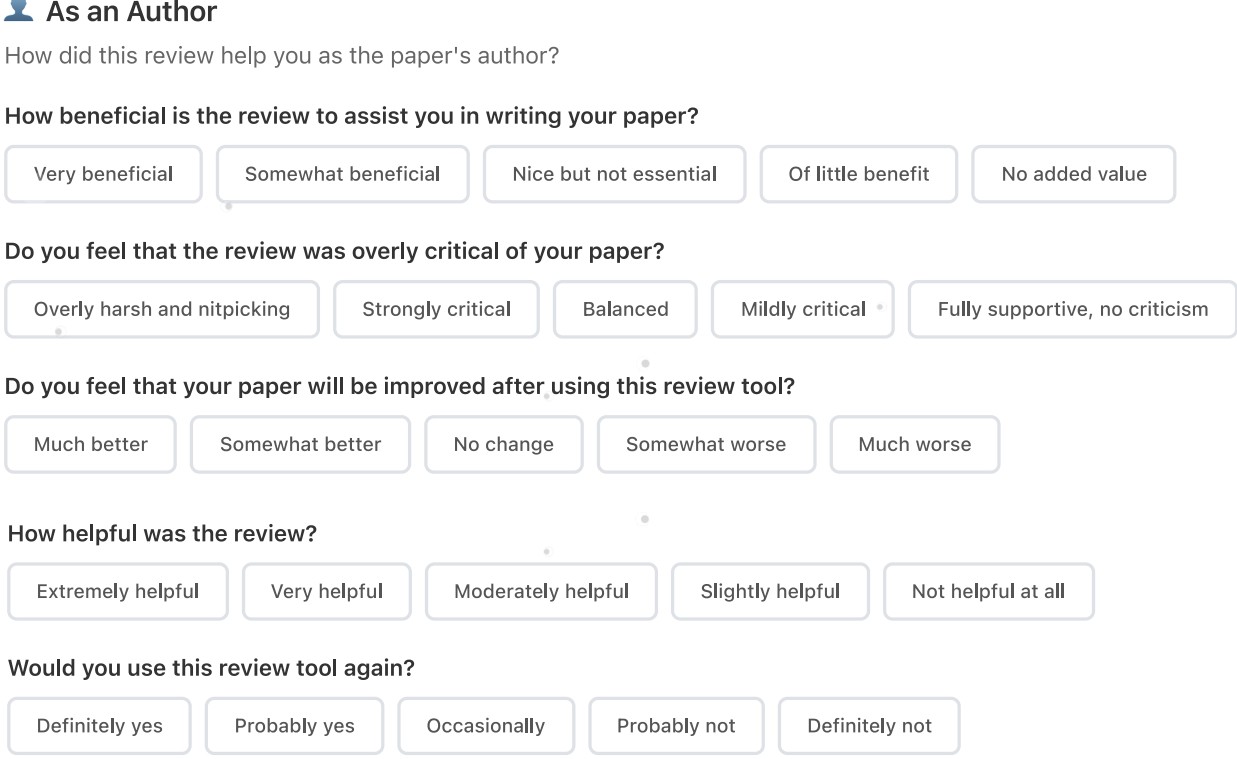

| Very beneficial | Somewhat beneficial | Nice but not essential | Of little benefit | No added value |

**Do you feel that the review was overly critical of your paper?**

| Overly harsh and nitpicking | Strongly critical | Balanced | Mildly critical | Fully supportive, no criticism |

**Do you feel that your paper will be improved after using this review tool?**

| Much better | Somewhat better | No change | Somewhat worse | Much worse |

**How helpful was the review?**

| Extremely helpful | Very helpful | Moderately helpful | Slightly helpful | Not helpful at all |

**Would you use this review tool again?**

| Definitely yes | Probably yes | Occasionally | Probably not | Definitely not |

Figure 25: Global survey questions and their 5-point Likert Scale asked at the end of the review. From top to bottom, each question corresponds to the metrics for *Beneficiality*, *Criticality*, *Improvement*, *Helpfulness*, and *Reuse Intention*.

**Questions** (from reviewers)

1. How do the token length differences between AI-generated and human-generated text (shown in Table 1) potentially bias the evaluation? Could detectors be inadvertently learning length-based features? 👍 ⬜ 👎

2. The rewriting experiments show particularly concerning results where none of the detectors show improvement[6] in TPR@0.01. Could you provide more analysis on why this scenario is so challenging for all detector types? 👍 ⬜ 👎

3. Given that output quality does not significantly impact the difficulty of detection[7], what factors do drive detectability differences across models and domains? 👍 ⬜ 👎

> **How accurate is the "Questions" section in analyzing your paper?**
>
> | Very accurate | Somewhat accurate | Insufficient information | Inaccurate | Completely misunderstood |

> **How generic is the "Questions" section in analyzing your paper?**
>
> | Highly specific and detailed | Paper-specific | Moderately specific | Somewhat generic | Entirely generic |

Figure 26: Example of section-level and comment-level feedback collected for the Questions section of the review. The same questions are asked for every section and comment throughout the review. The top and bottom questions assess the *Accuracy* and *Specificity* of the section respectively, while the thumbs up icon represents agreement with the comment, the gray square represents a neutral stance, and the thumbs down represents disagreement.

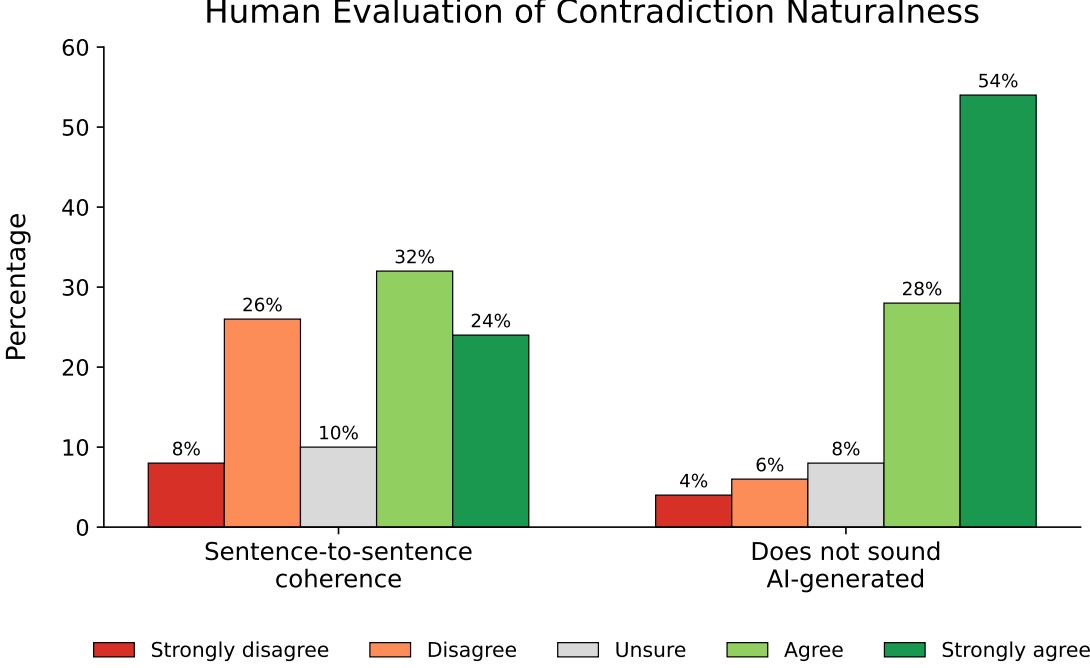

Figure 27: Human evaluation results on whether the generated contradictions read naturally in the sense of sentence-to-sentence coherence and without obvious signs of AI-generation. Raters evaluated each contradiction on a 5-point Likert scale from strongly disagree to strongly agree. We report the percentage of contradictions rated for each scale point.

# B    Additional Experimental Results

In this section, we present additional experimental results relating to the Contradiction Benchmark, WithdrarXiv-Check dataset, conference submissions, and focus-level evaluations.

## B.1    Contradiction Benchmark

**Synthetic Contradictions May Not Resemble Genuine Errors**    In Figure 27, we present the results of a small manual audit on 50 contradictions generated by GPT-4.1 in our benchmark. We ask human raters to determine if the contradictions read naturally along two dimensions: if they flow fluently from one sentence to the next, and if they avoid obvious signs of AI-generated phrasing. The contradictions are evaluated on a 5-point Likert scale ranging from strongly disagree to strongly agree, and we report the percentage of contradictions assigned to each response category.

From the figure, we observe that human raters judged 34% of the artificially inserted contradictions as not flowing naturally within the surrounding text. On the other hand, only 8% of the contradictions sound obviously AI-generated. These results align with our expectation that synthetic contradictions, even when manually inserted, may not always resemble genuine mistakes. This artifact may make them easier to detect, not because they appear obviously AI-generated, but because they disrupt the natural flow of the surrounding text. However, as shown in Figure 3, review systems already struggle to detect these comparatively easier errors, highlighting a weakness in current systems. We therefore view this benchmark as a first step toward verification-centric reviewing and aim to improve the naturalness of the errors in future work.

Table 14 reports a detailed breakdown of the accuracy of each LLM review system on our proposed Contradiction Benchmark, categorized by node distance. Recall that the node distance should be inversely correlated with the severity of a contradiction, where the severity indicates how strongly a review will be negatively impacted by the contradiction. Included in the table is the overall accuracy of each system on

Table 14: Accuracy of each system on the Contradiction Benchmark for various node distances from 0-8. We include our method, MLR, for ensembles of 4 reviews, a single review, and with the three-prompt chain combined into one (MLR-Combined). LLM-Review (Claude) refers to an ablation in which LLM-Review uses Claude Sonnet 4 without truncation. The last column (**Full**) is the average of the entire dataset. The best results are in **bold**; second-best values are underlined.

| Method | Node Distance | | | | | | | | | Full |
|--------|------|------|------|------|------|------|------|------|------|------|
| | 0 | 1 | 2 | 3 | 4 | 5 | 6 | 7 | 8 | **Full** |
| MLR (1 review) | 60.79 | 32.43 | 19.56 | 14.47 | 12.85 | 4.17 | 16.67 | 0.00 | 0.00 | 28.32 |
| MLR (4 reviews) | **73.43** | **47.17** | **31.67** | **27.41** | **21.46** | **15.21** | **38.89** | 0.00 | 0.00 | **40.95** |
| LLM-Review (Claude) | 35.40 | 22.75 | 7.77 | 7.68 | 6.99 | 8.33 | 0.00 | 0.00 | 0.00 | 16.43 |
| LLM-Review | 14.56 | 7.89 | 3.63 | 3.03 | 2.20 | 2.29 | 0.00 | 0.00 | 0.00 | 6.39 |
| AI Reviewer | 11.17 | 7.73 | 6.89 | 3.73 | 1.71 | 2.08 | 0.00 | **14.00** | 0.00 | 6.50 |
| AgentReview | 14.81 | 6.33 | 3.82 | 2.98 | 1.30 | 0.00 | 0.00 | 0.00 | 0.00 | 5.95 |
| MLR-Combined | 56.67 | 31.42 | 18.03 | 9.89 | 3.77 | 1.43 | 0.00 | 0.00 | 0.00 | 24.81 |

the whole dataset (**Full** column) as well as results of MLR when combining all three passes into a single prompt. Further discussion on the combined prompt setting can be found in Appendix B.4. Figure 28 shows the performance of our MLR system on the benchmark when aggregating 1 to 6 reviews, broken down by node distance as well. From our additional experiments, we conclude two main insights.

**Decoupling Model Choice and System Design**   There are two key factors that differentiate our MLR system from the others: i) our choice of LLM, using Anthropic's Claude instead of OpenAI's GPT; ii) our "multi-layered" system design driven by paper comprehension. While Table 14 and Figure 3 demonstrate a substantial performance advantage over all baselines, the extent to which each factor contributes to this improvement is uncertain. We aim to disambiguate this by performing an ablation on the system design. Consequently, we swapped the GPT model in the LLM-Review system with the exact Claude model that we use in our MLR system (`claude-sonnet-4-20250514`) and evaluated it on the Contradiction Benchmark. For this experiment, we only generated one review per paper and we use LLM-Review as it is the simplest and most cost-effective. We refer to this configuration as "LLM-Review (Claude)" in the table.

As observed in Table 14, simply by changing the model, there is already a significant improvement in contradiction detection, especially in node distances 0 and 1, the most crucial types of contradictions. This finding suggests the importance of optimizing the foundation LLM used in any agentic system. Further comparing LLM-Review (Claude) with MLR (1 review), there is another substantial performance gain in accuracy, across node distances 0-4, and over the full dataset. For contradictions with node distance 0, the accuracy increases by about 25% and 12% overall, a slightly higher increase than that from LLM-Review to LLM-Review (Claude). Considering both improvements suggests that each factor plays a vital role in enhancing the error detection abilities of an LLM reviewer. However, we note that, due to Claude's high unit cost, incorporating it into the other baseline systems may become relatively expensive, highlighting an important advantage of our MLR system (Table 9).

**Ablation on Review Ensembles**   As the AI Reviewer and AgentReview use ensembles of 5 and 3 reviews respectively, for a fair comparison, we report our results on the benchmark after aggregating 4 reviews. To fully examine the effect of varying the number of review ensembles, we conduct additional experiments using between 1 and 6 reviews and plot our results for node distances 0 to 6 in Figure 28. In the plot, we observe that the largest increase in accuracy across most node distances occurs when ensembling 2 reviews instead of 1. Moreover, increasing the number of ensembles continues to improve the system's performance in detecting contradictions, without plateauing. This trend suggests that the system benefits from diversity, highlighting the opportunity for future research to leverage ensemble advantages while minimizing cost.

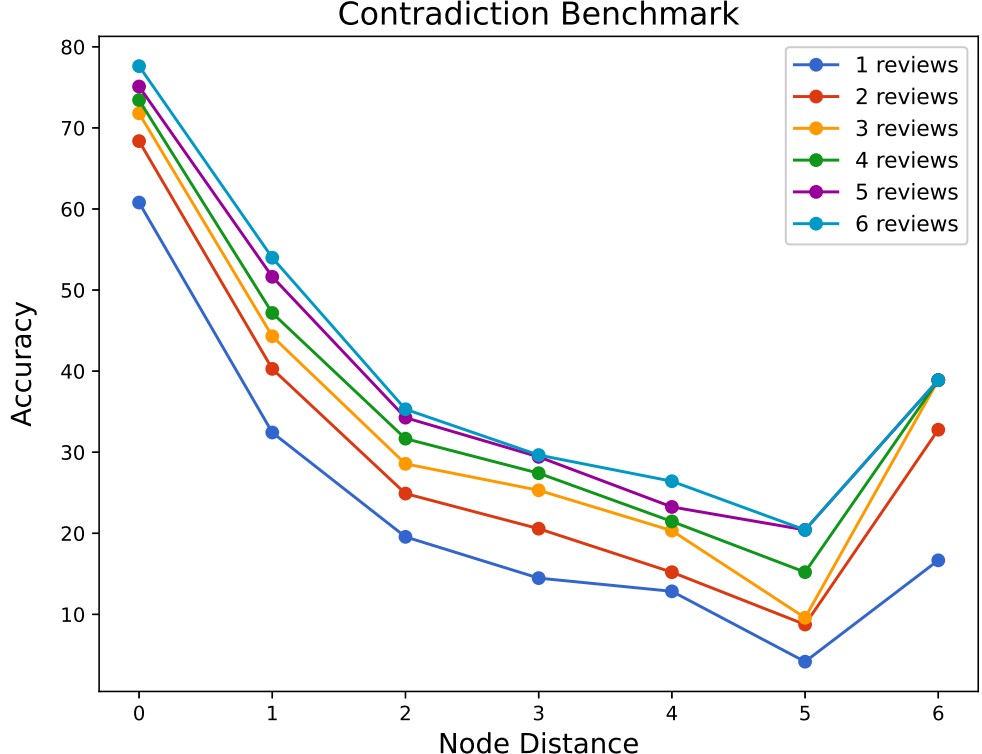

Figure 28: Accuracy of our MLR system on the Contradiction Benchmark, plotted by node distance (0–6) for ensembles of 1–6 reviews.

Table 15: Breakdown of MLR's accuracy (%) on the WithdrarXiv-Check dataset by page count (range is inclusive) and subject for both settings in the LLM-judge. "Similar" captures cases where the review raises concerns comparable to the retraction comment, whereas "Exact" refers to exact matches. Subjects with fewer than 11 papers are grouped under "Others". Accuracies are computed relative to the number of papers in each group. For example, MLR detected 35.1% of errors on "math" papers with 1-10 pages and 24.8% of all "math" papers under the "Similar" setting.

| Page count/ | 1–10 | | | 11–20 | | | 21–30 | | | Total | | |
|---|---|---|---|---|---|---|---|---|---|---|---|---|
| Subject | # Paper | Similar | Exact | # Paper | Similar | Exact | # Paper | Similar | Exact | # Paper | Similar | Exact |
| math | 37 | 35.1 | 21.6 | 46 | 21.7 | 17.4 | 22 | 13.6 | 9.1 | 105 | 24.8 | 17.1 |
| cs | 9 | 66.7 | 22.2 | 15 | 26.7 | 13.3 | 7 | 14.3 | 14.3 | 31 | 35.5 | 16.1 |
| physics | 10 | 40.0 | 20.0 | 5 | 80.0 | 80.0 | 1 | 0.0 | 0.0 | 16 | 50.0 | 37.5 |
| cond-mat[5] | 10 | 0.0 | 0.0 | 2 | 50.0 | 0.0 | 3 | 33.3 | 0.0 | 15 | 13.3 | 0.0 |
| Others | 24 | 25.0 | 16.7 | 11 | 9.1 | 0.0 | 9 | 11.1 | 11.1 | 44 | 18.2 | 11.4 |
| Total | 90 | 32.2 | 17.8 | 79 | 25.3 | 17.7 | 42 | 14.3 | 9.5 | 211 | 26.1 | 16.1 |

## B.2 WithdrarXiv-Check Dataset

Table 15 provides a detailed breakdown of MLR's results on the WithdrarXiv-Check dataset, allowing us to further understand its modest overall performance. We group papers by subject and page count range, using 10 pages bins. For each group and setting ("Similar" and "Exact"), we report the number of papers together with their accuracies relative to the number of papers in the group. Subjects with fewer than 11 papers are grouped under "Others" for readability.

---

[5]Condensed Matter

We observe that under the "Exact" setting, both computer science (cs) and math papers have roughly the same accuracies of around 16–17%, while in the "Similar" setting, MLR detects a higher percentage of errors in cs papers at 35.5% compared to math at 24.8%. Furthermore, accuracy tends to decrease as paper length increases. These findings suggest that both domain and context length limits contribute to MLR's moderate performance on WithdrarXiv-Check relative to the Contradiction Benchmark. As mentioned in Section 2.3, another reason is that contradictions in the benchmark have stylistic artifacts that make them easier to detect than genuine errors.

## B.3 Focus-level Evaluations

In Figures 29–32, we present radar plots of the four focus distributions (strength-target, strength-aspect, weakness-target, weakness-aspect) of LLMs and review systems as in Table 5, excluding those already found in Section 4.3. An examination of the figures reveals two key insights.

**Strength: LLMs provide more specific feedback and OpenAI models match human emphasis on Communication Clarity.** From Figures 29 and 30, we observe that human reviewers allocate a substantial proportion of their strength-related comments to the Paper facet, which captures general remarks about the manuscript. Their feedback also places greater emphasis on communication clarity. In contrast, LLM reviewers provide fewer general comments, offer more specific feedback, and devote relatively less attention to communication clarity. Notably, OpenAI models (GPT-5 mini, GPT-5.1, GPT-4o mini) deviate from this trend by exhibiting an emphasis on Communication Clarity that is closely aligned with human reviewers.

A plausible explanation is that human reviewers, under limited time and high reviewing workload, often avoid exhaustive verification of technical details. As a result, they place greater emphasis on communication clarity as a proxy for overall paper quality: a manuscript that is easier to read, well-structured, and clearly articulated reduces the effort required to assess its contributions and is therefore perceived more favorably. This naturally leads to a higher proportion of strength-related comments focused on clarity and general high-level remarks. However, this emphasis may become less critical in practice, as modern LLMs can effectively support authors in improving clarity and presentation, reducing the effort required to produce well-articulated manuscripts, thereby improving the effectiveness of LLM-based reviewing with different focuses.

**Weakness: Gemini models focus more on Method and human reviewers criticize Novelty more heavily than LLM reviewers.** From the weakness target distribution in Figure 31, we observe a pronounced peak in the Method facet for the Gemini models (Gemini 2.5 Pro and Gemini 3 Pro), indicating a strong and distinctive emphasis on methodological weaknesses, rather than experimental or motivational ones. This may reflect Gemini's preference for text-grounded critiques that can be justified with high confidence since evaluating experimental design or motivation often requires substantial contextual and domain-specific judgment. From this perspective, such behavior can be viewed as a desirable property, favoring precision and reliability in automated reviews over unsubstantiated extrapolations. In contrast, the weakness aspect distribution in Figure 32 shows that human reviewers are more critical of novelty than LLM reviewers. This pattern suggests that human reviewers place greater weight on original contributions as part of their acceptance criteria, consistent with the goal of advancing academic research. LLMs appear to lack this perspective, possibly due to the narrow scope defined in the system prompt, which focuses solely on reviewing the paper and does not provide broader context regarding how paper acceptance contributes to the growth of knowledge in the field. A possible avenue for improvement in the future.

## B.4 Evaluation on Conference Submissions

Building on the experiments conducted in Section 4.4, we use the same set of papers for further analysis.

**Three-Prompt Chain vs. Combined Prompt** Prior work shows that breaking tasks into multi-step reasoning prompts improves accuracy on structured reasoning tasks, reduces hallucination, and enhances factuality through iterative or verifiable intermediate steps (Wei et al., 2022; Yao et al., 2023; Xu et al.,

Figure 29: Radar plot of the strength target focus distribution for human reviewers, review systems, and LLMs not featured in the main text.

2025). Accordingly, the three-prompt chain in our MLR system enables us to assess and optimize each guided pass the LLM takes over the paper. However, we notice that since every pass must include the previous passes and the full paper again as input, this leads to considerable token usage. While prompt chaining is useful for detailed refinement of each stage in the review generation process, considerations of token efficiency motivate evaluating a version of the system in which the three finalized prompts are combined into a single prompt.

We present the Pearson, Spearman, and Kendall's Tau correlation for our system with the combined prompt in Table 16 as MLR-Combined and include the other baselines, as seen in Table 7 for reference. These are correlations between scores predicted by an LLM-judge and actual human review scores. As observed in the table, MLR-Combined's correlations are of similar value to our original MLR system. In Table 14, we also include the accuracy of MLR's combined prompt variant on a subset (497 papers) of the Contradiction Benchmark and find a small drop in performance of around 3.5% overall when using 1 review. These results indicate that there would not be a substantial difference in concatenating all three prompts into one. Furthermore, the meaningful impact of our system is due to its design in explicitly guiding the model towards a deep understanding of the paper before producing the review.

Figure 30: Radar plot of the strength aspect focus distribution for human reviewers, review systems, and LLMs not featured in the main text.

**Correlations by Decision Outcome** In Table 17, we provide a more granular breakdown of the correlation scores by decision outcome. The results corroborate those in Table 7 with LLM-Review showing a strong performance on NeurIPS 2024 papers, but not on any other venues. Our MLR system and the AI Reviewer accordingly do relatively well on all conferences, though in some cases (e.g., ICLR 2025), MLR attains higher overall correlation, whereas the AI Reviewer performs better when analyzed by decision. This shows that while our system excels at differentiating paper quality overall, the AI Reviewer has stronger sensitivity for finer-grained assessment within decision categories.

**Correlation Analysis of System Scores** Since LLM-Review does not inherently output a score for each paper, we use an LLM-judge for score predictions to ensure comparable results between all baseline review systems. As a complementary analysis, we include a correlation evaluation using each system's generated review scores, if available. These are presented in Table 18. Regarding NeurIPS 2024 papers, the AI Reviewer had a much stronger correlation when using their generated scores than the LLM-judge's predictions, achieving the best performance among the other systems. A possible reason for this could be the system's use of the full NeurIPS review template, for both written and numeric sections, thereby guiding the model toward scores with closer alignment to human reviewers' judgments under the same criteria. In line with this reasoning, the AI Reviewer's generated scores have considerably poorer correlation for other

Figure 31: Radar plot of the weakness target focus distribution for human reviewers, review systems, and LLMs not featured in the main text.

venues, as compared with the LLM-judge's predicted scores. These results highlight the strong influence of review templates on LLM reviewers, particularly in calibrating consistency with human reviewers. In contrast, because our MLR system employs a conference-agnostic review template, the scores it generates actually correlate with human reviewers as well as, or even better than, the predictions. This is despite using the same NeurIPS scoring rubrics as the AI Reviewer.

**Low Variance of Scores Generated by AgentReview** Figure 33 displays a scatterplot of the review scores generated by AgentReview for each conference, averaged across each reviewer. The colors of the markers indicate if the score was generated for an accepted or rejected paper and the horizontal lines represent the average score within each decision outcome. Recall that their scoring rubrics closely follows ICLR. As evidenced by the proximity of the average lines to each other and the interspersed red and green scatter with no clear separation, AgentReview's scoring system is unable to distinguish between papers that meet acceptance standards and those that do not. Their scores are mainly within a small interval, between 4 and 6, and have limited spread. A likely reason for the lack of high scores is due to the instructions provided to reviewers:

> *"Do not assign scores of 7 or higher before the rebuttal unless the paper demonstrates exceptional originality and significantly advances the state-of-the-art in machine learning."*

Figure 32: Radar plot of the weakness aspect focus distribution for human reviewers, review systems, and LLMs not featured in the main text.

However, as there were no explicit instructions to prevent low scores, their occurrence may be attributed to insufficient calibration. The inherently low variance in scores may explain the small score difference observed under explicit manipulation.

Table 16: Pearson, Spearman, and Kendall's Tau correlation between the predicted scores of model-generated and human reviews across venues. Included in gray as a reference is the LLM-judge's prediction of paper scores based on the human reviews. MLR-Combined is our proposed review system with the three-prompt chain combined into one. Highest values, excluding the human baseline, are **bolded** while second highest values are underlined.

| Venue | Method | Pearson | Spearman | Kendall |
|-------|--------|---------|----------|---------|
| NeurIPS 2024 | Human (Reference) | 0.781 | 0.700 | 0.547 |
| | MLR-Combined | 0.395 | 0.355 | 0.282 |
| | MLR | **0.451** | 0.386 | 0.299 |
| | LLM-Review | 0.358 | **0.457** | **0.377** |
| | AI Reviewer | 0.328 | 0.331 | 0.265 |
| | AgentReview | 0.167 | 0.139 | 0.103 |
| ICML 2025 | Human (Reference) | 0.684 | 0.599 | 0.477 |
| | MLR-Combined | 0.417 | 0.335 | 0.291 |
| | MLR | 0.429 | 0.333 | 0.277 |
| | LLM-Review | 0.169 | 0.081 | 0.070 |
| | AI Reviewer | **0.439** | **0.416** | **0.353** |
| | AgentReview | 0.006 | 0.054 | 0.049 |
| ICLR 2025 | Human (Reference) | 0.742 | 0.754 | 0.577 |
| | MLR-Combined | **0.606** | 0.548 | 0.448 |
| | MLR | 0.586 | **0.574** | **0.472** |
| | LLM-Review | -0.013 | -0.006 | -0.003 |
| | AI Reviewer | 0.538 | 0.453 | 0.369 |
| | AgentReview | 0.195 | 0.204 | 0.172 |

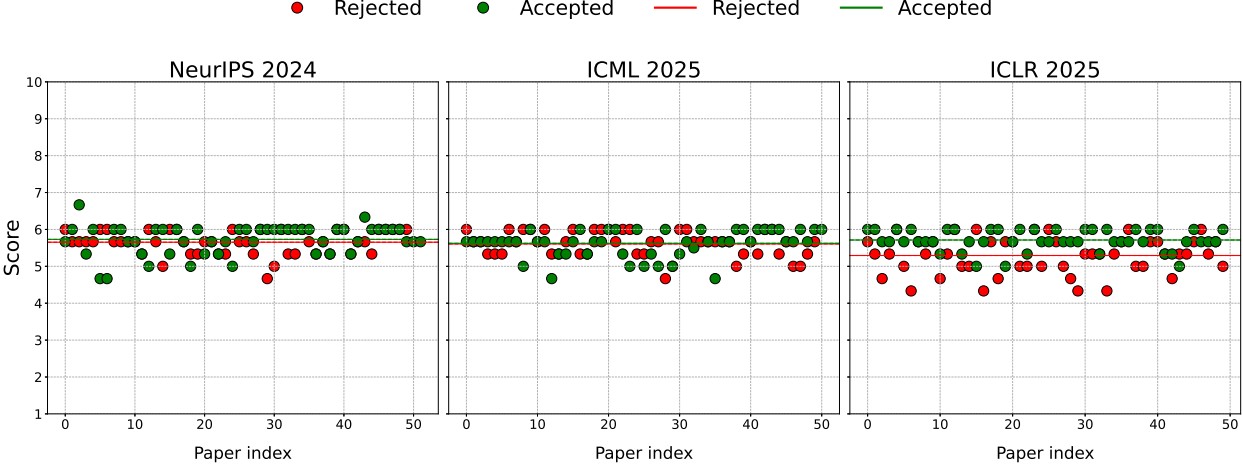

Figure 33: Scatterplot of review scores by AgentReview on papers from NeurIPS 2024, ICML 2025, and ICLR 2025. Red markers and lines represent rejected papers while green ones represent accepted papers. The red and green horizontal lines are the average scores within each category of decision outcome, while the markers are the scores themselves.

Table 17: Pearson, Spearman, and Kendall's Tau correlation between the predicted scores of model-generated and human reviews, split by acceptance outcome and venue. Included in gray as reference are the LLM-judge's predicted scores based on human reviews. MLR-Combined is our proposed review system with the three-prompt chain combined into one. Highest values, excluding the human baseline, are **bolded** while second highest values are underlined.

| Decision | Venue | Method | Pearson | Spearman | Kendall |
|---|---|---|---|---|---|
| | | Human (Reference) | 0.677 | 0.657 | 0.497 |
| | | MLR-Combined | 0.042 | 0.032 | 0.026 |
| | NeurIPS 2024 | MLR | 0.060 | 0.038 | 0.020 |
| | | LLM-Review | **0.195** | **0.220** | **0.185** |
| | | AI Reviewer | -0.004 | -0.012 | -0.013 |
| | | AgentReview | 0.049 | 0.019 | 0.008 |
| | | Human (Reference) | 0.477 | 0.482 | 0.376 |
| | | MLR-Combined | 0.118 | 0.098 | 0.085 |
| Accepted | ICML 2025 | MLR | 0.116 | 0.116 | 0.099 |
| | | LLM-Review | 0.067 | 0.026 | 0.023 |
| | | AI Reviewer | **0.305** | **0.277** | **0.241** |
| | | AgentReview | -0.086 | -0.009 | -0.004 |
| | | Human (Reference) | 0.163 | 0.235 | 0.173 |
| | | MLR-Combined | **0.167** | 0.109 | 0.092 |
| | ICLR 2025 | MLR | 0.060 | 0.071 | 0.064 |
| | | LLM-Review | -0.186 | -0.171 | -0.145 |
| | | AI Reviewer | 0.147 | **0.134** | **0.115** |
| | | AgentReview | -0.298 | -0.208 | -0.165 |
| | | Human (Reference) | 0.733 | 0.661 | 0.510 |
| | | MLR-Combined | 0.267 | 0.175 | 0.140 |
| | NeurIPS 2024 | MLR | 0.279 | 0.195 | 0.154 |
| | | LLM-Review | **0.408** | **0.381** | **0.315** |
| | | AI Reviewer | 0.196 | 0.211 | 0.169 |
| | | AgentReview | 0.090 | 0.048 | 0.031 |
| | | Human (Reference) | 0.703 | 0.601 | 0.484 |
| | | MLR-Combined | **0.437** | 0.377 | 0.333 |
| Rejected | ICML 2025 | MLR | 0.394 | 0.296 | 0.243 |
| | | LLM-Review | 0.235 | 0.138 | 0.119 |
| | | AI Reviewer | 0.403 | **0.397** | **0.338** |
| | | AgentReview | -0.109 | -0.066 | -0.049 |
| | | Human (Reference) | 0.666 | 0.675 | 0.511 |
| | | MLR-Combined | 0.470 | 0.356 | 0.289 |
| | ICLR 2025 | MLR | 0.367 | 0.349 | 0.282 |
| | | LLM-Review | 0.174 | 0.175 | 0.149 |
| | | AI Reviewer | **0.558** | **0.505** | **0.409** |
| | | AgentReview | 0.398 | 0.341 | 0.282 |

Table 18: Pearson, Spearman, and Kendall's Tau correlation between scores from review systems themselves and human reviews across conferences. MLR-Combined is our proposed review system with the three-prompt chain combined into one. The best result per conference is highlighted in **bold** and the second best is underlined.

| Venue | Method | Pearson | Spearman | Kendall |
|---|---|---|---|---|
| NeurIPS 2024 | MLR-Combined | 0.406 | 0.363 | 0.297 |
| | MLR | 0.432 | 0.384 | 0.311 |
| | AI Reviewer | **0.445** | **0.444** | **0.350** |
| | AgentReview | 0.066 | 0.083 | 0.064 |
| ICML 2025 | MLR-Combined | **0.517** | **0.386** | **0.333** |
| | MLR | 0.445 | 0.305 | 0.258 |
| | AI Reviewer | 0.374 | 0.311 | 0.250 |
| | AgentReview | 0.134 | 0.073 | 0.056 |
| ICLR 2025 | MLR-Combined | **0.668** | 0.671 | 0.554 |
| | MLR | 0.660 | **0.693** | **0.563** |
| | AI Reviewer | 0.460 | 0.383 | 0.300 |
| | AgentReview | 0.391 | 0.336 | 0.257 |

