# OpenReview forum: "Beyond Imitation: A Framework and Benchmark for LLM-Assisted Peer Review"
_TMLR — Decision pending for TMLR_

### Review · Reviewer_q7iV · 2026-05-16

**Summary Of Contributions:**

Contributions
1. The paper introduces a verification-centric paradigm for AI-assisted peer review, emphasizing contradiction and error detection rather than imitation of human-written reviews.
2. It proposes a Contradiction Benchmark together with a knowledge-graph-based severity evaluation framework for systematically assessing review systems.
3. It presents a Multi-Layered Review (MLR) framework that performs progressively deeper manuscript understanding before generating reviews.

Strengths
1. The problem formulation is novel and highly relevant to practical peer review settings.
2. The proposed benchmark is meaningful and enables quantitative evaluation of contradiction detection ability.
3. The experimental evaluation is comprehensive, including withdrawn-paper analysis, reviewer focus analysis, and human evaluation.

Weaknesses
1. Although the authors include human evaluations, much of the large-scale (automatic) evaluation still relies on LLM as judges, which may introduce evaluation bias.
2. It is not entirely clear whether the observed improvements come from the proposed framework itself or from increased token usage and stronger underlying models.

**Audience:**

Yes

**Audience Explanation:**

Yes. I think this paper studies a timely and increasingly important problem, especially given the growing use of LLMs in scientific workflows and peer review. The proposed benchmark and verification-focused perspective feel meaningful beyond this specific task, and I believe many researchers in LLM agents, AI for science, and peer review systems would find the findings interesting.

**Broader Impact Concerns:**

I do not have significant broader impact concerns. The authors already acknowledge potential misuse risks and discuss appropriate safeguards and human oversight.

**Claims And Evidence:**

Yes

**Claims Explanation:**

Yes. The paper provides comprehensive experimental evidence, including benchmark evaluations, robustness studies, and human evaluations, which together support the main claims convincingly and clearly.

**Requested Changes:**

It would strengthen the paper to include some additional discussions or comparisons clarifying whether the performance gains primarily come from the proposed MLR framework itself, or from (increased) token usage. In general, stronger models and larger token budgets often lead to better performance.

---

> ### Author Response · Authors · 2026-06-02
> **Response to review**
>
> We thank the reviewer for their thoughtful and constructive comments, and greatly appreciate the effort taken to review our work carefully. Below, we address your concerns in turn and have colored all our changes in blue in the revised manuscript:
>
> **Weaknesses:**
>
> 1. We acknowledge the reviewer’s concern regarding potential evaluation bias from the use of LLM-judges. At the same time, many of these evaluations require a large number of data points to be empirically meaningful, making fully manual evaluation impractical. In addition, several of the evaluation protocols are adopted from prior work, which we follow to ensure consistency and comparability. For our own proposed evaluations, we take additional steps to increase confidence in the protocol, including human verification, as reported in Appendix A.1. We will also add an experiment that manually assesses the sensitivity of the LLM-judge (pending).
>
> 2. A common concern raised by all reviewers is determining the extent to which the observed performance gap on the Contradiction Benchmark is driven by the choice of the underlying model versus the design of our MLR system. The ablations reported in Table 14 directly address this point and should be presented in the main text rather than the appendix to avoid giving readers a misleading impression. Accordingly, we have added an additional line plot to Figure 3 to reflect these results, along with a short discussion in Section 4.1 and a minor revision to the abstract.
>
> 	From Table 9, we also observe that for most experiments in the paper, the token budget for MLR is smaller ( ~200K) as compared with the AI Reviewer ( ~400K) and AgentReview ( ~300K) due to review ensembling in the latter systems. Therefore, we do not attribute MLR’s improved performance to an increase in token budget.
>
> We hope that we have sufficiently addressed the reviewer’s concerns and believe that these changes have helped to strengthen our manuscript. We look forward to a fruitful discussion!

---

> > ### Comment · Reviewer_q7iV · 2026-06-06
> > **Thanks authors for the response**
> >
> > Thanks for your response! It makes much sense to me and I have no other concerns.

---

> ### Author Response · Authors · 2026-06-06
> **Thanks to the reviewer and update on response**
>
> Thank you for your acknowledgement, we are glad to have addressed your concerns!
>
> Additionally, we would like to include an experiment that manually assesses the sensitivity of the LLM-judge as mentioned before. From the reviews generated by MLR, we identified 50 reviews where the contradiction was successfully detected and o3 should return the answer "yes". To verify the accuracy of o3, we found that, over 10 runs per paper, the LLM-judge has a sensitivity of 86.8%. This suggests that o3 modestly underestimates each review system’s accuracy on our benchmark and the reported accuracies are likely to be slightly conservative.
>
> We have also included this result in our analysis in Appendix A.1 in the revised manuscript.

---

### Review · Reviewer_Y8WA · 2026-05-19

**Summary Of Contributions:**

This paper makes contributions in two main ways: First, they propose a way to evaluate LLM peer review systems. Second, they propose their own LLM peer review system, designed to focus on verifying the correctness of a paper’s claims.

Strengths/contributions:
* The paper proposes a systematic way of introducing contradictions in articles at varying levels of severity, based on a constructed knowledge graph.
* The authors create an LLM-based review system (“Multi Layered Review”) with an Appendix Agent, Literature Review Agent, and Review Agent. The Appendix Agent (small model, Claude Haiku 3.5) summarizes the Appendix. The Literature Review Agent (Claude Sonnet 4 with websearch tool) generates a literature review. The outputs of both of these agents are then handed off to the Review Agent (Claude Sonnet 4), who then reads the paper three times: The first time, it produces an outline. In the second pass, it finds supporting evidence for key points and looks for weaknesses. In the third pass, it incorporates information from the Appendix Agent and Literature Review agent to produce a final review with Strengths, Weaknesses, Questions, Overall Recommendation, and Score.
* They conduct evaluations on a dataset of withdrawn papers (WithdraXiv-Check), a dataset of ML conference papers, and user studies with real researchers to determine if it provides useful reviews.
* They demonstrate that their agent focuses more on validity compared to human reviewers, providing complementarity instead of aiming to replicate human behaviors.

Weaknesses:
* On realistic errors (from WithdraXiv-Check), the performance of MLR is better than baselines but still not so good (23.7%, Table 3)
* The MLR system is not well equipped to handle longer papers due to context window limitations
* Unclear if MLR is good at identifying mistakes in proofs, which is a particularly promising avenue for creating LLM agents that complement human reviewers.
* Unclear how much of the lift of MLR relative to other LLM review systems is due to system design choices vs. choice of underlying LLM.

**Audience:**

Yes

**Audience Explanation:**

LLM agents are certainly of broad interest, and since many people are already using LLMs for paper reviewing, it is worthwhile to consider systematic ways of incorporating LLM agents into the reviewing pipeline and how we should design evaluate such agents.

**Broader Impact Concerns:**

A Broader Impact Statement is included in the discussion section, and it points out the risk of reviewers relying on such LLM systems and leading them to not fully engage with the paper.

**Claims And Evidence:**

Yes

**Claims Explanation:**

Yes, the paper performs extensive experiments on simulated and real data.

**Requested Changes:**

1. Can you provide examples in the appendix of what an error at each node level looks like?
2. The proposed framework relies on prompt-defined agents so the prompts should be provided for reproducibility purposes, unless there is a compelling reason not to. In particular, I am curious whether the Appendix Agent is designed to detect errors in proofs.
3. Can you report the total cost of experiments for the paper?
4. Figure 10: Bar charts should be avoided for visualizing non-monotonic metrics. Figures (+ caption) should ideally be interpretable on their own, and it is not clear that it is better for Criticality to be close to neutral, instead of maxed out at 5 (as with the other metrics). Alternative options include a dot, with standard errors, or a box and whisker plot.
5. Figure 12 is very hard to parse. Can you try visualizing this information in a different way?
6. If the goal is to champion the MLR method, it would be more honest to include the ablations in Table 12 in the main paper instead of deferring them to the appendix, because it should be argued that the success of MLR compared to baselines is attributable to the system design rather than simply using a better underlying LLM.
7. This is a minor thing, but I don’t really think the title of the paper appropriately conveys the unique contributions of the paper. Perhaps it should include "verification" or something like that, but this is not critical.

Small corrections:
* bottom of p. 41: “openAI” -> “OpenAI”
* numbers <= should be spelled out (e.g., “five” instead of 5)
* The references should be cleaned up. The capitalization is not correct for many entries.

---

> ### Author Response · Authors · 2026-06-02
> **Response to review**
>
> We thank the reviewer for their thoughtful and constructive comments, and greatly appreciate the effort taken to review our work carefully. Below, we address your concerns in turn and have colored all our changes in blue in the revised manuscript:
>
> **Weaknesses:**
>
> We agree with the reviewer’s first two points: MLR achieves only modest performance on realistic errors, and its performance is limited in part by context-length constraints. At the same time, MLR was developed primarily to support the review of AI conference papers, which are typically limited to around 10 pages and fall within the computer science domain. The majority of the WithdrarXiv-Check dataset consists of mathematics papers. Thus, although MLR outperforms the other review systems, its remaining limitations highlight areas for future improvement, including extending the system to broader domains.
>
> To further analyze the reasons for MLR’s performance, we have included a breakdown of MLR’s results on the WithdrarXiv-Check dataset by subject and page count in Appendix B.2 along with a short discussion. The results in Table 3 will also change slightly, as the batch output file from the LLM-judge in the previous experiment was not saved before it expired. The revised breakdown is computed using the same review files from all review systems, with only the LLM-judge evaluation repeated.
>
> In summary, we find that there are three reasons for the modest results:
>
> 	i) The main reason is likely that genuine errors are more difficult to detect than the synthetic errors introduced into each paper in our benchmark. We discuss this further in Section 2.3.
>
> 	ii) Domain limits also play a role as we observe lower accuracy on computer science (cs) papers, which we consider AI-related, than on math papers.
>
> 	iii) Context length limitations are another factor as we find that accuracy decreases as page count increases.
>
> Next, regarding MLR’s ability to identify mistakes in proofs, the results do not show clear evidence of such capability. On the WithdrarXiv-Check dataset, about half are math papers and we achieve an accuracy of 24.8% (17.1%) in the Similar (Exact) setting. Furthermore, when filtering for retraction comments containing the word “proof”, MLR’s accuracy is 15.2% and 8.7% in the Similar and Exact settings, respectively.
>
> However, we recognize the importance and practical value of proof verification in reviewing, and leave its further development to future work.
>
> **Requested changes:**
> 1. We have included examples of errors at node levels 0-8 in Figures 14, 15, and 16.
>
> 2. We agree with the reviewer that the manuscript should provide sufficient detail to ensure reproducibility. Therefore, we have added the prompts used for the MLR system in Figures 18, 19, 20, and 21 of the appendix. As seen in Figure 18, the Appendix Agent is simple and does not contain any explicit instructions to check for errors in proofs.
>
> 3. A loose estimate of the total API cost for all experiments in the paper is approximately 3,500 USD. This is mainly because we use batching and flex processing for most experiments.
>
> 4-5. Yes, we agree that Figures 10 and 12 are not the best way to visualize these results. Instead, we have omitted Figure 10 and simply reported values in the discussion and converted Figure 12 into Table 10 in the revised manuscript. This should improve the clarity and readability of the paper.
>
> 6. A common concern raised by all reviewers is determining the extent to which the observed performance gap on the Contradiction Benchmark is driven by the choice of the underlying model versus the design of our MLR system. We agree that the ablations reported in Table 14, which directly address this point, should be presented in the main text rather than the appendix to avoid giving readers a misleading impression. Accordingly, we have added an additional line plot to Figure 3 to reflect these results, along with a short discussion in Section 4.1 and a minor revision to the abstract.
>
> 7. Regarding the title, we agree that the original title may sound somewhat generic. We are considering the revised title “Beyond Imitation: A Framework and Verification-Centric Benchmark for LLM-Assisted Peer Review,” while also taking into account the overall length of the title.
>
> Lastly, the small corrections have largely been addressed, except for the numbers that should be spelled out. In Section 4.1, we considered the ensemble of reviews and distances of nodes to be like hyperparameters, which is why we use numbers there. We would be grateful if the reviewer could point us to the parts requiring correction, and we would be happy to change them.
>
> We hope that we have sufficiently addressed the reviewer’s concerns and believe that these changes have helped to strengthen our manuscript. We look forward to a fruitful discussion!

---

> > ### Comment · Reviewer_Y8WA · 2026-06-12
> > **Response to authors**
> >
> > I thank the authors for the effort they put into incorporating reviewer feedback. The updated manuscript is a clearer and more honest presentation of the work, which I believe will be of interest to the TMLR audience. Regarding not spelling out numbers that are hyperparameters -- this makes sense. My one remaining comment is the last column of the newly added Table 3 is redundant, and removing it would improve clarity.

---

> > > ### Author Response · Authors · 2026-06-12
> > > **Thank you for your response**
> > >
> > > Thank you for your response. We are glad to have addressed your concerns, and we believe that your comments have helped us strengthen the manuscript.
> > >
> > > May we clarify if you are referring to the last column in Table 3, or possibly another table (like Table 12)? As the last column of Table 3 reports results of one of the baselines.

---

> > > > ### Comment · Reviewer_Y8WA · 2026-06-12
> > > > **Correction**
> > > >
> > > > Apologies, I meant Table 10 (the Disagreed Comments column)

---

> > > > > ### Author Response · Authors · 2026-06-12
> > > > > **Revised Table 10**
> > > > >
> > > > > Thank you for the clarification! We have revised Table 10 accordingly.

---

### Review · Reviewer_AwqS · 2026-05-26

**Summary Of Contributions:**

The paper makes three contributions to the study of LLM-assisted peer review. First, it introduces a Contradiction Benchmark of 1,164 data points constructed by inserting synthetic errors into 257 papers from ACL, AISTATS, CVPR, ICML, and NeurIPS. The novelty here is the use of a paper-specific knowledge graph (generated with Gemini 2.5 Pro under a constrained schema) to assign each contradiction a severity proxy based on its shortest-path distance from a "main claim" node. The authors validate this severity calibration empirically by showing that detection accuracy declines monotonically with node distance for all evaluated systems.

Second, the authors propose the Multi-Layered Review (MLR) framework, a three-agent system with an Appendix Agent (Claude Haiku 3.5), a Literature Review Agent (Claude Sonnet 4 with web search), and a Review Agent (Claude Sonnet 4) that executes a three-pass prompt chain over the main text inspired by Keshav's three-pass reading approach. The design choice worth highlighting is the priority on deep manuscript comprehension before review generation, rather than ensembling or chain-of-thought.

Third, the paper provides a fairly comprehensive evaluation against three baselines (LLM-Review, AI Reviewer, AgentReview) across six axes: contradiction detection, WithdrarXiv-Check retraction detection, focus distribution against the Shin et al. framework, score correlation on ICLR/ICML/NeurIPS submissions, explicit prompt-injection robustness, cost, and a small (N=38) user study.

The headline empirical findings are: MLR detects contradictions at substantially higher rates than baselines (about 73% at distance 0 versus 11-15% for baselines with 4-review ensembles); it correlates with human review scores at Pearson 0.43-0.59 across venues; it produces reviews whose focus distribution diverges from human reviewers more than simpler LLM baselines do (average KL 0.240, the highest in Table 5); and it incurs about USD 0.50 per review excluding the optional literature review.

Key strengths in my view are the knowledge-graph severity calibration, the breadth of the evaluation suite, and the honest reporting of weaknesses (the cost discussion, the high std dev on manipulation robustness, the explicit acknowledgement that the system is vulnerable to adversarial editing). The main weaknesses are that part of the performance gap appears attributable to model choice rather than system design (the ablation in Appendix B.1 makes this clear but the main-text framing does not), the synthetic-contradiction benchmark lacks human validation of error quality, and the user study lacks a control condition.

**Additional Comments:**

This is a solid applied-systems paper. The comprehensiveness of the evaluation is the main strength; few papers in this space evaluate across six independent axes (contradiction detection, retraction detection, focus distribution, score correlation, manipulation robustness, cost) and report a user study on top. The honest reporting of weaknesses, including the cost limitations, the explicit-manipulation vulnerabilities and the results, is the kind of writing that makes a paper trustworthy to read.

The framing of the contribution as "augment rather than replace" human reviewers is well-chosen and the verification-centric perspective is the right reorientation for this subfield. I expect the Contradiction Benchmark to be useful to other researchers working in this area independent of whether MLR itself becomes a widely adopted system.

Thank you to the authors for the work. I look forward to the response.

**Audience:**

Yes

**Audience Explanation:**

The TMLR audience includes researchers working on LLM evaluation, peer-review reform, and applied LLM systems for scientific workflows. The paper's contribution is relevant to all three groups.

The Contradiction Benchmark fills a gap that researchers working on automated review have been circling around for some time. Existing efforts (WithdrarXiv, the Skarlinski et al. contradiction work) are either retraction-driven and therefore vague, or rely on prompt-extracted errors rather than evaluating end-to-end review systems. A benchmark of 1,164 contradictions with paper-specific severity calibration, built by an automatable pipeline that can be extended to new conferences, is a useful artifact even if the specific MLR system in this paper is later superseded.

The focus-distribution analysis in Section 4.3 is independently interesting. The finding that LLM reviewers underweight communication clarity in strengths and underweight novelty in weaknesses relative to human reviewers is the kind of empirical observation that has design implications for anyone building review-assistant tools. The expanded set of models compared to Shin et al. (adding GPT-5 mini, GPT-5.1, Gemini 2.5 Pro, Gemini 3 Pro, Claude Sonnet 4.5) makes this section a useful reference point on its own.

The cost analysis in Table 9 is the kind of transparent reporting that TMLR's audience benefits from. Most papers in this area do not report per-component token costs, which makes it hard to evaluate practical deployment claims.

The user study, despite its small N, contains a genuinely useful finding: the weak negative correlation between Criticality and both Helpfulness and Reuse Intention. This runs counter to the intuition (built up from human peer review) that rigor and critique are synonymous, and it points to a real design question for assistive LLM tools.

The paper would also be of interest to readers thinking about adversarial robustness of LLM systems more broadly, since the explicit-manipulation section provides evidence that all four evaluated review systems remain vulnerable to a fairly straightforward prompt-injection attack, with detection rates that are nonzero only for MLR.

I would not recommend this paper to readers primarily interested in fundamental LLM capabilities research, since it is an applied systems paper. But for the applied-LLM and scientific-workflow audience that TMLR serves, the findings are of practical and conceptual value.

**Broader Impact Concerns:**

The existing Broader Impact Statement addresses how reviewers might rely too heavily on the system and appropriately emphasizes human oversight. One dynamic not currently covered is the author side: the same capability that helps authors strengthen submissions can also be used to iteratively refine papers against automated review checks. This is not a misuse case so much as a structural tension — the system's value as a reviewer assistant and its value as an author assistant are partially in conflict, since both depend on the same error-detection capability. A brief mention of this dynamic would round out the statement.

**Claims And Evidence:**

Yes

**Claims Explanation:**

The central claims of the paper are supported by the evidence presented, though several are stronger when read against the appendix than against the main text. I work through them in turn.

The claim that MLR achieves substantially higher contradiction detection than baselines is well supported by Figure 3 and Table 12, with Wilson confidence intervals reported for small bins. The benchmark itself is large enough (1,164 contradictions across five venues) to support cross-system comparisons, and the inverse relationship between node distance and detection accuracy across all four systems is consistent evidence that the severity calibration captures something real about contradiction importance.

The claim that the system is "venue-agnostic within the main machine learning conferences" is supported by Table 7 and the box plots in Figure 9, where MLR maintains comparable score-correlation patterns across NeurIPS 2024, ICML 2025, and ICLR 2025. I would not extend this claim beyond ML conferences, and the authors do not, but the within-ML evidence is reasonable.

The cost-efficiency claim is supported by Table 9, which is unusually transparent about per-component token counts and unit prices. The Appendix B.3 finding that the combined-prompt variant achieves comparable performance at roughly one-third the cost is a useful piece of evidence that the three-pass design is not strictly necessary for the reported gains.

Two claims deserve more careful scrutiny.

First, the framing that MLR's gains stem from the "multi-layered" system design needs the ablation in Appendix B.1 (Table 12) to be promoted to the main text. Swapping LLM-Review's underlying model from GPT-4.1 to Claude Sonnet 4 raises its single-review accuracy on the Contradiction Benchmark from 6.39% to 16.43% overall, and from 14.56% to 35.40% at distance 0. MLR with a single review reaches 28.32% overall and 60.79% at distance 0. So the model swap accounts for roughly 40% of the absolute single-review gap at distance 0, with the remaining gain attributable to system design. This is a real and meaningful system-design contribution, but the main text reads as if the design alone is responsible. The appendix is clear about this; the abstract and Section 4.1 should be too.

Second, the claim of "comparatively greater robustness" to explicit manipulation in Section 4.5 rests on Table 8's mean score change of 0.26 for MLR (Self) versus 1.32 (AI Reviewer) and 0.43 (AgentReview). The authors are admirably honest that "all systems are strongly influenced by the injected text" and that std devs exceed 2 for both LLM-judge and Self scores. Given that variance, a mean delta of 0.26 with a standard deviation above 2 on N=50 papers is not strong evidence of a robustness difference. The 8/50 detection cases (where MLR flagged the injection) are the more meaningful robustness result and deserve more emphasis than the score-delta comparison. I would also note that the std devs in Table 8 are reported on the raw before/after scores, not on the per-paper deltas, which is the quantity that actually matters for assessing the robustness claim.

The LLM-judge methodology is reasonable and the authors validate o3 carefully in Appendix A.1 with a 99.9% accuracy on unmodified papers (where the correct answer is always "no"). This validates the judge's specificity but not its sensitivity, since the harder failure mode for the benchmark is false negatives on actual contradictions, which would inflate every system's reported numbers similarly but obscure absolute performance. The two qualitative examples comparing o3, GPT-4.1, and o4-mini in Appendix A.1 are helpful for confidence in the judge's behavior on edge cases.

The human evaluation findings (Section 4.7) are correctly hedged. With N=38 sessions and no control condition, the correlations between Criticality and Helpfulness (r = -0.29) are suggestive rather than confirmatory, and the authors describe them as such.

Overall the evidence supports the paper's core claims when read carefully, with the qualifications above.

**Requested Changes:**

The changes below are organized from those I consider important for acceptance to those that would strengthen the work.

**Important for acceptance:**

1. The robustness claim in Section 4.5 needs to be reworked. The current text argues that MLR has "comparatively greater robustness" to explicit manipulation based on a mean score change of 0.26, but with standard deviations exceeding 2 on N=50, this is not statistically distinguishable from the other systems. The before/after standard deviations are also reported separately rather than on per-paper deltas, which is the quantity that actually matters for the claim. The 8/50 detection cases where MLR flagged the injection are the more defensible robustness signal. Either compute per-paper deltas with their dispersion, or lead with the detection result and treat the score deltas as secondary evidence.

2. The synthetic-contradiction methodology needs a limitations discussion. Some contradictions generated by GPT-4.1 may carry stylistic or structural artifacts that make them easier to detect than naturally occurring errors, particularly for LLM-based review systems that may pick up on the same generation patterns. A small manual audit (say, 50 contradictions) with human raters assessing whether the inserted error reads naturally in context would strengthen confidence in the benchmark. If such an audit has been done, it should be reported.

3. The configuration of the MLR system varies across evaluations: the Literature Review Agent is excluded from the Contradiction Benchmark, WithdrarXiv-Check, and explicit manipulation experiments; both Literature Review and Appendix Agents are excluded from focus distribution and conference submission evaluations; the full system is used only in the user study. This is noted in Appendix A.2, but the variation should be stated at each relevant results section so readers understand what configuration produced the headline numbers — particularly when those numbers are compared against baselines that have no equivalent optional components.

**Would strengthen the work:**

4. The LLM-judge validation in Appendix A.1 establishes o3's specificity on unmodified papers (correctly returning "no" 99.9% of the time) but not its sensitivity on actual contradictions. A small human-validated subset where the ground-truth answer is known to be "yes" would let readers gauge whether the reported detection rates are over- or under-estimates.

5. The WithdrarXiv-Check results in Table 3 are modest in absolute terms (14-24% detection even under the "Similar" relaxation). The authors attribute this to domain mismatch (math/physics journals versus ML conferences), which is plausible. A breakdown of MLR's performance on the math/physics subset versus any ML-adjacent papers in the dataset would help readers understand whether the system generalizes outside ML or whether the modest results reflect domain limits.

6. Appendix B.3 shows the combined-prompt variant achieves comparable score correlation to the three-pass chain at roughly one-third the cost, and the authors themselves conclude there is no substantial difference. However, this comparison is only on score correlation, not on contradiction detection, which is the paper's headline metric. Running the combined-prompt variant on the Contradiction Benchmark would clarify whether the three-pass structure contributes meaningfully to error detection or whether it is primarily a cost overhead.

7. In Section 2.1.2, the prompt in Figure 14 allows both "Supports" and "Contradicts" relationships, but the authors note that "Contradicts" is never used by the model in practice. This raises a question about whether the schema is doing useful work as designed, or whether removing "Contradicts" would change the resulting graph structure at all. A small test would resolve this.

---

> ### Author Response · Authors · 2026-06-02
> **Response to review (1/2)**
>
> We thank the reviewer for their detailed and constructive comments, and greatly appreciate the effort taken to review our work carefully. Below we address your concerns in turn and have colored all our changes in blue in the revised manuscript:
>
> 1. **Performance gap due to underlying model versus system design**
>
> 	A common concern raised by all reviewers is determining the extent to which the observed performance gap on the Contradiction Benchmark is driven by the choice of the underlying model versus the design of our MLR system. We agree that the ablations reported in Table 14, which directly address this point, should be presented in the main text rather than the appendix to avoid giving readers a misleading impression. Accordingly, we have added an additional line plot to Figure 3 to reflect these results, along with a short discussion in Section 4.1 and a minor revision to the abstract.
>
> **Important for acceptance:**
>
> 2. **Robustness claim in Section 4.5**
>
> 	We acknowledge that due to the large standard deviation and small sample size, the claim of “stronger comparative robustness” should not be attributed to the score change, and instead is better evidenced by the 8/50 detection cases. We have revised the results paragraph in Section 4.5 to address this point and included the per paper delta standard deviations in Table 8.
>
> 3. **Synthetic-contradiction benchmark lacks human validation of error quality**
>
> 	We would like to thank the reviewer for bringing this point to our attention. Indeed, it makes sense that due to the synthetic nature of the contradictions, the errors introduced are easier to detect than genuine mistakes. This could also be a reason for the modest results on the WithdrarXiv-Check dataset and we mention this in our results discussion in Section 4.2. We have included a limitations section (Section 2.3) to discuss this as well as included the results of a manual audit that we did on 50 contradictions in Appendix B.1, as suggested by the reviewer. During the manual audit, we asked human raters if the contradictions sounded natural in the sense of coherence from sentence-to-sentence as well as if they sounded human-written. Raters were given a 5-point Likert scale from strongly disagree to strongly agree for evaluation.
>
> 	We found that human raters judged 34% of the contradictions as not flowing naturally from sentence to sentence while only 8% were judged to sound AI-generated. These results echo our intuition that the artificially introduced errors have structural artifacts that enable easier detection. However, we observe that review systems still struggle to detect these comparatively easier errors, suggesting a limitation of current systems. We therefore view this benchmark as an initial step toward verification-centric reviewing.
>
> 4. **Including the configuration of the MLR system at each relevant result section**
>
> 	The revised manuscript now specifies the MLR system configuration used to produce each result in the corresponding sections, improving the reproducibility of our work. For clarity, we added a table summarizing the settings, along with each baseline review system’s corresponding MLR agent configuration in Appendix A.2, Table 11. The same baseline settings are used for all evaluations.
>
> 	Additionally, we would like to apologize for a genuine mistake in our appendix and correct it. Inadvertently, two sentences were swapped in the manuscript, leading to a misunderstanding of our experimental settings.
>
> 	The original paragraph reads (mistake):
> 	>''Specifically, we exclude a literature review in our analyses on the Contradiction Benchmark, WithdrarXiv-Check dataset, and explicit manipulation in submissions. Regarding the evaluations on conference submissions and focus distributions, we simply allow the review agent to perform a web search, instead of explicitly calling for a literature review to reduce costs. **As these evaluations are not necessarily affected by the contents of the appendix, to further minimize cost, we also do not include the appendix agent.**''
>
> 	The revision reads (corrected):
>
> 	>''Specifically, we exclude a literature review in our analyses on the Contradiction Benchmark, WithdrarXiv-Check dataset, and explicit manipulation in submissions. **As these evaluations are not necessarily affected by the contents of the appendix, to further minimize cost, we also do not include the Appendix agent.** Regarding the evaluations on conference submissions and focus distributions, we simply allow the Review agent to perform a web search, instead of explicitly calling for a literature review to save costs as well.''

---

> > ### Author Response · Authors · 2026-06-02
> > **Response to review (2/2)**
> >
> > **Would strengthen the work:**
> >
> > 5. **LLM-judge validation (pending)**
> >
> > Indeed, we agree with the reviewer that including an additional experiment on a small subset of papers where the ground truth answer is “yes” (i.e. on papers where the review generated by a system has caught the contradiction) would improve the validity of our LLM-judge. We are currently still working on it and will include it in the next reply and final manuscript. Though, from the specificity (correctly returning “no”), we observe that o3 has a low false positive rate, implying that the performance of each review system is not over-estimated. While the sensitivity helps us estimate the extent to which the LLM-judge underestimates their performance.
> >
> > 6. **Breakdown of WithdrarXiv-Check results**
> >
> > We have included a breakdown of MLR’s results on the WithdrarXiv-Check dataset by subject and page count in Appendix B.2 along with a short discussion. The results in Table 3 will also change slightly, as the batch output file from the LLM-judge in the previous experiment was not saved before it expired. The revised breakdown is computed using the same review files from all review systems, with only the LLM-judge evaluation repeated.
> >
> > In summary, we find that there are three reasons for the modest results:
> >
> > 	i) The main reason is likely that genuine errors are more difficult to detect than the synthetic errors introduced into each paper in our benchmark.
> >
> > 	ii) Domain limits also play a role as we observe lower accuracy on computer science (cs) papers, which we consider AI-related, than on math papers.
> >
> > 	iii) Context length limitations are another factor as we find that accuracy decreases as page count increases.
> >
> > 7. **Combined-prompt variant on Contradiction Benchmark**
> >
> > Added into the Appendix B.1 Table 14 are the results of running MLR’s combined prompt variant on a subset of the Contradiction Benchmark (497 papers). We use this subset to limit costs, and believe it is sufficiently large to support our analysis. The overall accuracy of MLR with the combined prompt is relatively similar to MLR (1 review) at 24.81% versus 28.32% respectively. We run both experiments using 1 review.
> >
> > From these findings, we can conclude that the three-pass structure has no substantial impact on both the error detection abilities nor score correlations with actual human reviews. However, we do note that from node distances 3 onwards there is a significant decrease in accuracy while performance on nodes 0-2 are much closer to the original MLR system. It is likely that the three-pass structure makes a more meaningful difference on mistakes that are difficult to detect.
> >
> > 8. **"Supports" and "Contradicts" relationships in Knowledge Graph (pending)**
> >
> > We agree with the reviewer that the “Contradicts” relationship likely has limited effect on knowledge graph formation, as the Gemini model appears unlikely to explicitly flag contradictions and the papers that we used to generate these graphs do not contain contradictions themselves. However, we have done some testing on papers after introducing our artificial contradiction, using the same prompt with the same schema, and there is still no use of the “Contradicts” relationship.
> >
> > We will explicitly test this by removing this relationship and assessing whether the graph changes meaningfully. Our findings will be reported in the next response.
> >
> > 9. **Broader impact statement**
> >
> > In the revised manuscript, we have included a brief mention of the impact of these review systems from the author’s side.
> >
> > We hope that we have sufficiently addressed the reviewer’s concerns and believe that these changes have helped to strengthen our manuscript. We look forward to a fruitful discussion!

---

> > > ### Author Response · Authors · 2026-06-06
> > > **Update on our response**
> > >
> > > We would like to provide a brief follow-up with additional results addressing the reviewer’s concern on:
> > >
> > > **Would strengthen the work:**
> > >
> > > 5. **LLM-judge validation**
> > >
> > > We include an experiment that manually assesses the sensitivity of the LLM-judge as suggested by the reviewer. From the reviews generated by MLR, we identified 50 reviews where the contradiction was successfully detected and o3 should return the answer "yes’". To verify the accuracy of o3, we found that, over 10 runs per paper, the LLM-judge has a sensitivity of 86.8%. This suggests that o3 modestly underestimates each review system’s accuracy on our benchmark and the reported accuracies are likely to be slightly conservative.
> > >
> > > We have also included this result in our analysis in Appendix A.1 in the revised manuscript.
> > >
> > > 8. **"Supports" and "Contradicts" relationships in Knowledge Graph**
> > >
> > > In our further analysis of the knowledge graphs, we realized that from the 259 graphs generated, there were 6,947 "Supports" relationships and 28 "Contradicts" relationships; this is equivalent to about 0.4% of relationships being "Contradicts". Hence, we would like to correct the overly strong statement in the manuscript that says "never" to "rarely". Our original analysis was based on manually inspecting approximately 30 graphs during the initial stage of designing our knowledge graph generation procedure.
> > >
> > > We had also used gemini-2.5-pro-preview-06-05 which has since been discontinued. Therefore, we run our knowledge graph generation procedure again with and without the "Contradicts" relationship using gemini-2.5-pro for comparison.
> > >
> > > Due to the variance in responses when using LLMs (using a temperature of 0.1 for our experiments), there will also be some differences in the generated knowledge graphs even if the same schema was provided. To account for this, we generated three graphs for each of 100 papers: two using a schema that includes the "Contradicts" relationship, and one using a schema without it. We will denote them as G_with1, G_with2, and G_without respectively.
> > >
> > > Then, for each node type, we count the average number of nodes that differ between G_with1 and G_with2 as well as between G_with1 and G_without. Our results are summarized in Table 1 below. As observed, the differences are largely within the usual variance of LLM outputs and the "Contradicts" relationship does not seem to have much impact on the construction of the knowledge graphs. However, since the "Contradicts" relationship appears in 7 out of 2,544 relationships in G_with1 and 8 out of 2,602 relationships in G_with2​, these few relationships would necessarily be affected by removing it from the schema.
> > >
> > > **Table 1: Average difference in number of nodes per node type between two generated knowledge graphs that includes the "Contradicts" relationship (G_with1 vs G_with2), and between one that includes and one that does not (G_with1 vs  G_without).**
> > > | Node type | G_with1 vs G_with2 | G_with1 vs  G_without |
> > > |---|---:|---:|
> > > | Mathematical_Theory | 0.50 | 0.49 |
> > > | Evidence | 1.13 | 0.96 |
> > > | Methodology | 1.34 | 1.42 |
> > > | Implementation | 0.84 | 0.85 |
> > > | Claim | 1.99 | 1.77 |

---

> > > > ### Comment · Reviewer_AwqS · 2026-06-08
> > > > **Response to authors**
> > > >
> > > > Thanks for the thorough response. My concerns are addressed.
> > > > The new Section 2.3 and the manual audit of the 50 contradictions is a useful addition — the 34% non-natural-flow finding is the kind of thing authors often hide, and reporting it openly makes the benchmark stronger. Same with the corrected configuration sentences in Appendix A.2.
> > > > The reframing of the Section 4.5 robustness claim toward the 8/50 detection cases is the right move. With the per-paper delta std devs you're now reporting, that's clearly where the signal lives.
> > > > One small note on the LLM-judge sensitivity: 86.8% is informative, but the test set was drawn from reviews where MLR had already detected the contradiction, so it's really sensitivity conditional on detection rather than across the population. A sentence in Appendix A.1 noting that scope would help readers calibrate.
> > > > On the three-pass ablation: honest result, and the distance 3+ carve-out is defensible. Worth making sure the abstract and intro framing catches up — the original positioning leaned heavily on the multi-pass design being central.
> > > > The "Contradicts" work in the follow-up is well-done. Correcting "never" to "rarely" with the actual count and the controlled removal experiment is the right way to handle it.
> > > > No remaining concerns.

---

> > > > > ### Author Response · Authors · 2026-06-12
> > > > > **Thanks for your response**
> > > > >
> > > > > Thank you for your response. We are glad to have addressed your concerns. Your comments have helped us strengthen the manuscript and it has been revised accordingly.

---

### Review · Reviewer_ZC66 · 2026-07-01

**Summary Of Contributions:**

This paper proposes two things
1) A synthetic "contradiction" dataset and benchmark created by adding false facts into a set of real AI conference papers. Contradicting claims are placed at varying distances from the main claim and paragraphs were rewritten to contradict the original point.

2) Several different LLM-assisted peer review designs to identify said errors. An "appendix agent", "lit review agent", and "review agent" work together in a 3 pass review process to identify contradictions and potential withdrawl.

3) Lastly, the benchmark score from each proposed review design is compared to human reviewers and tested for ancillary metrics like prompt injection robustness and cost.

The main strength is the benchmark itself - it identifies a narrowly-scoped problem of contradiction detection that an LLM can help a human reviewer with. Then it creates a benchmark to test whether several different designs of LLM review systems can catch these errors.

The main weaknesses that the system design claims are harder to isolate than the benchmark contribution. For example, gains appear to come from a mixture of model choice, context handling, structure, prompting, and system design. Specifically, there is an ablation where a better model using a combined prompt achieved similar results to the 3-pass architecture, suggesting that the architecture of the LLM review itself may not be the most impactful or sole cause of the measured performance.

**Audience:**

Yes

**Audience Explanation:**

Benchmark: yes.
Multi-agent reviewer: not as useful because LLMs themselves have improved greatly. I think the findings would be more interesting if the paper were rerun with a modern set of models.

**Broader Impact Concerns:**

The prompt injection experiment raises important concerns, albeit tangential. This seems like it could be a serious risk, and a follow-up experiment could be done to determine if there is an additional LLM review system design that can better identify or guard against prompt injection.

**Claims And Evidence:**

Yes

**Claims Explanation:**

The main claim is supported:
- The paper makes a good case that LLM review systems should be evaluated on whether they can catch errors, and not just whether the reviews look like human reviews.
- The benchmark and resulting system performance is also clear.

The claim that MLR performs better is not fully supported. Some of the gain comes from using a stronger underlying model. LLM review seems to perform much better with a different model. If the claim is that the three-pass design is the best performer, I think more experimentation or analysis of agent traces and logs should be done.

The WithdrarXiv-Check experiment is useful, but the results are modest. MLR catches about 26% of errors under the relaxed setting and 16% under the exact setting. It is unclear what is needed to achieve a better detection score.

**Requested Changes:**

I would like to see better isolation of each confounding factor that contributes to performance.

At a minimum, please run the following:
- AI Reviewer + Claude
- AgentReview + Claude
- MLR + GPT-4.1/o4-mini/GPT-4o
- All systems with a common model
It's unclear why AI Reviewer uses o4-mini, LLM-Review uses GPT-4.1, AgentReview uses GPT-4o, while MLR uses Claude.

A clear analysis of LLM reviewer-only failure modes would provide a much clearer justification of the various review system designs.

---

### Decision · Action_Editor_Faxw · 2026-07-08

**Recommendation:** Accept with minor revision

**Additional Comments:**

I recommend acceptance with minor revision. The final version should ensure that the abstract and introduction do not overstate the role of the multi-layered architecture, clearly state the scope of the LLM-judge sensitivity result, and consistently reflect the revised ablations and limitations.

**Audience:**

Yes

**Audience Explanation:**

The paper is relevant to TMLR readers interested in LLM evaluation, scientific workflows, peer-review assistance, and verification-oriented LLM systems. The benchmark and verification-centric framing are timely and useful.

**Claims And Evidence:**

Yes

**Claims Explanation:**

The main claims are supported by the experiments and additional analyses. The authors addressed concerns about model-vs-system effects, benchmark validity, robustness, LLM-judge reliability, and configuration clarity. The original reviewers indicated that their concerns were addressed.